# DNA mechanocapsules for programmable piconewton responsive drug delivery

Arventh Velusamy[1], Radhika Sharma[1], Sk Aysha Rashid[1], Hiroaki Ogasawara[1] & Khalid Salaita[1,2] ✉

The mechanical dysregulation of cells is associated with a number of disease states, that spans from fibrosis to tumorigenesis. Hence, it is highly desirable to develop strategies to deliver drugs based on the "mechanical phenotype" of a cell. To achieve this goal, we report the development of DNA mechanocapsules (DMC) comprised of DNA tetrahedrons that are force responsive. Modeling shows the trajectory of force-induced DMC rupture and predicts how applied force spatial position and orientation tunes the force-response threshold. DMCs functionalized with adhesion ligands mechanically denature in vitro as a result of cell receptor forces. DMCs are designed to encapsulate macromolecular cargos such as dextran and oligonucleotide drugs with minimal cargo leakage and high nuclease resistance. Force-induced release and uptake of DMC cargo is validated using flow cytometry. Finally, we demonstrate force-induced mRNA knockdown of HIF-1α in a manner that is dependent on the magnitude of cellular traction forces. These results show that DMCs can be effectively used to target biophysical phenotypes which may find useful applications in immunology and cancer biology.

A powerful strategy to increase the efficacy of drugs is to specifically deliver therapeutics to diseased cells. For example, antibody–drug conjugates are used to treat different types of cancer by targeting cell surface receptors to concentrate a cytotoxic drug at its intended target[1]. A complementary approach for reducing off-target effects and increasing drug efficacy involves creating inactive drugs that are activated upon encountering a unique stimulus in diseased tissue[2]. The most common strategy for creating such smart or triggered drugs involves unmasking the therapeutic using *chemical inputs* such as pH[3,4], redox state[5], small molecules[6], enzymatic activity[7,8], nucleic acids[9], and proteins[10].

We therefore devised a strategy to release and activate drugs using *mechanical inputs* generated by cell surface receptors. The motivation for using a specific magnitude mechanical force as a cue comes from quantitative measurements of forces generated by many classes of receptors such as Integrins[11,12], T cell[13,14], and B cell receptors[15,16], and Notch[17,18] among others[19,20]. The mechanical forces generated by cells and transmitted to their cell surface receptors is important to

biological processes such as signaling[13,18], migration[21,22], cell cycle progression[23,24], as well as giving cells and tissues their intrinsic shape and architecture[25]. In principle, mechano-targeting can augment the advances in the area of targeted delivery[26,27]. Moreover, mechanical dysregulation can indicate a diseased state even when cells have a biochemical composition and morphology identical to those of healthy cells[26].

Leveraging the altered mechanics of diseased cells as a trigger for drug deployment could, therefore, allow one to target diseased cells effectively. Given the well-characterized force responses of DNA nanostructures using single molecule force spectroscopy and modeling[28], we designed tetrahedral DNA (TD) structures to function as DNA mechanocapsules (DMCs). The choice of DMC scaffold was made considering the TD's efficient assembly and high tailorability[29], coupled with extensive work demonstrating in vitro and in vivo therapeutic delivery of small molecules[30], nucleic acids[31,32], and proteins[33,34]. Hence DMC could serve as a modular vehicle for encapsulated or interwoven cargos which could be delivered at programmable piconewton forces.

[1]Department of Chemistry, Emory University, Atlanta, GA, USA. [2]Wallace H. Coulter Department of Biomedical Engineering, Georgia Institute of Technology and Emory University, Atlanta, GA, USA. ✉e-mail: k.salaita@emory.edu

Here we show a DNA mechanocapsule that expands the toolset for precision drug delivery. We targeted integrin adhesion receptors as they are broadly expressed in adherent cells and transmit 10–50 pN forces to their ligands in a cell and tissue-specific manner[35]. We also describe the design and synthesis of drug-containing DMCs and the characterization of their stability and force-induced release. As a final proof-of-concept, we demonstrate force-triggered knockdown of HIF-1α using an antisense oligonucleotide that is currently in clinical testing as an anti-cancer therapeutic[36].

## Results

### Rational design of tetrahedral DNA mechanocapsules

We developed DMCs that deform upon experiencing pN forces and release their encapsulated contents (Fig. 1a). DMCs displayed integrin ligands (RGD) at specific sites such that pN forces from integrin receptors lead to significant structural disruption. The DMCs were comprised of 5–6 oligonucleotides that included chemical modification points for RGD conjugation, surface anchoring, and sites for fluorophore and quencher pairs (Fig. 1c). The RGD and anchor sites were orientated outward for binding while fluorophore and quencher attachment points were orientated inward shielding them within the DMC (Supplementary Fig. 2)[33]. Initially, three chemically identical DMCs but differing in RGD attachment points and force-bearing (FB) strand lengths were modeled to simulate DMC responses under forces

using oxDNA2 (Supplementary Fig. 3). Ideally, we would employ single molecule force spectroscopy techniques to validate DMC force response, but these measurements are challenging for large libraries of structures; moreover, oxDNA rupture force predictions have been validated and shown to be accurate when compared to experimental measurements[37]. The ligand site on the DMCs was pulled at a rate of $1.4 \times 10^4$ nm/s along the z-axis to estimate the force of rupture for these DMCs (Fig. 1b, Supplementary Videos 1–3). We found that DMCs initially oriented and deformed along the force axis, followed by gradual dehybridization and complete rupture of the FB strand (red strand) as the force ramped up. The plot in Fig. 1b shows the trajectory of DMC$_{39pN}$ that undergoes significant deformation prior to the release of the FB strand at 39 pN followed by a force drop due to rupture. Note that rupture force is highly loading rate dependent, as has been well documented in the literature[38,39], and lower loading rates will lead to dampened rupture force values. The loading rate used for DMC simulations was $1.4 \times 10^4$ nm/s, which was found to be appropriate as we bench-marked the rupture of dsDNA using oxDNA against that of force spectroscopy values and we see general agreement, thus validating the simulations (Supplementary Fig. 3G, H). Using this loading rate, modeling showed that the three DMCs displayed rupture forces of $27.0 \pm 0.6$ pN, $39.0 \pm 0.5$ pN, and $43.5 \pm 0.5$ pN (Fig. 1c).

To further demonstrate the tunability of DMCs, we engineered a structure that is non-responsive to cellular forces. We achieved this by

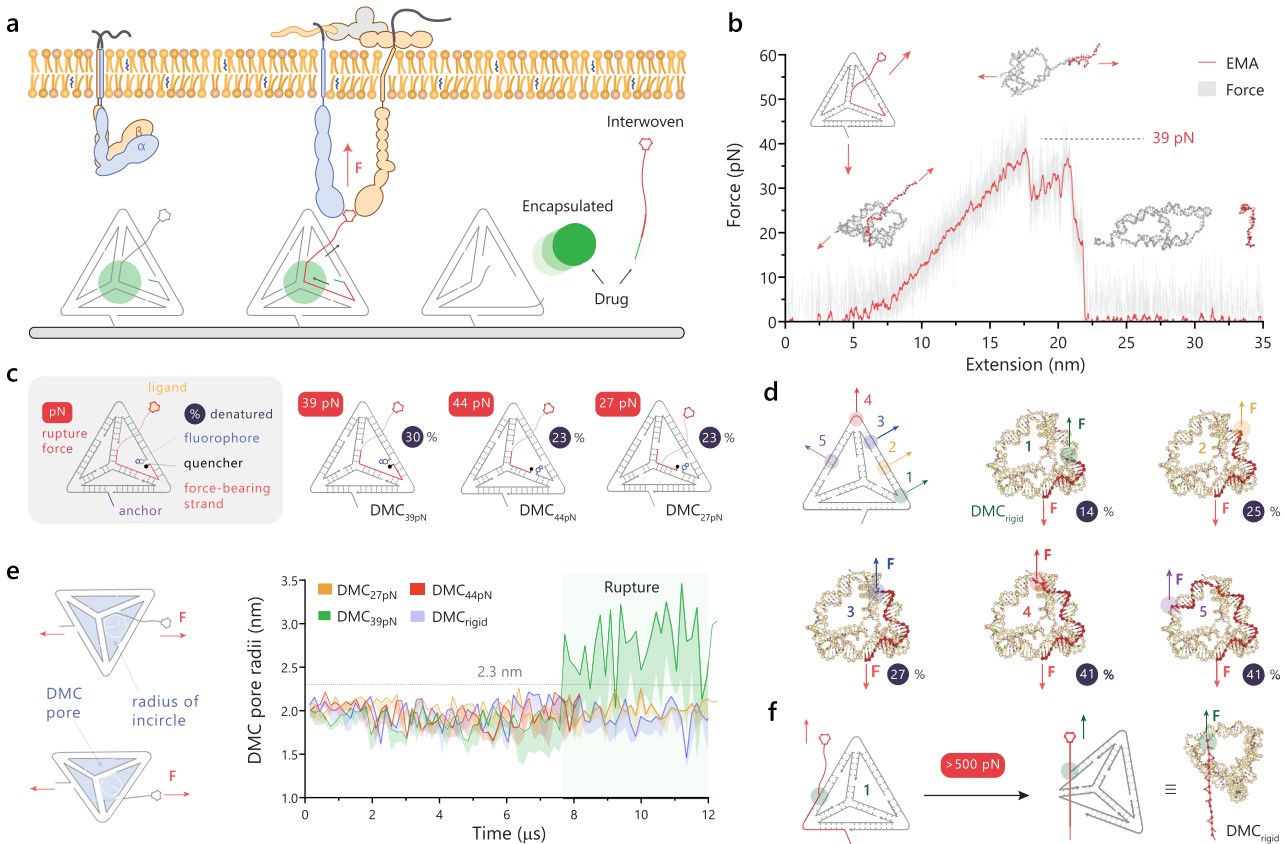

**Fig. 1 | oxDNA modeling of DMCs to predict unfolding. a** Schematic of cellular force-induced cargo release from DMC structures. **b** Force−extension curves of DMC$_{39pN}$ under a loading rate of $1.4 \times 10^4$ nm s$^{-1}$ generated using oxDNA. The exponential moving average (EMA) of the data points is denoted by a red line, and the gray line shows raw data points. The force was applied at the attachment points indicated by red arrows. The peak force of 39 pN is indicated by the dashed line. **c** Schematic labeling the different nucleic acid modifications in the DMC. The force values indicated in red show the peak rupture force from oxDNA simulations, while the value in the blue circle shows the percentage of base pairs that were mechanically denatured at the end of the simulation. **d** Library of force

non-responsive DMCs (DMC1−5) tested with ligands at different points on the anchoring strand and modeled using oxDNA. The external force was applied to the termini of the red strand and the number inside solid blue circle indicates the percentage of base-pairs denatured at the end of the simulation. **e** Plot of pore radii of DMC$_{27pN}$, DMC$_{39pN}$, DMC$_{44pN}$, DMC$_{rigid}$ when subjected to a force ramp of $1.4 \times 10^4$ nm s$^{-1}$. The solid lines indicate the maximum of the pore radii and the region between the maximum and mean radii are shaded. **f** Structure deformation of DMC$_{rigid}$ when subjected to forces as high as 500 pN in oxDNA. Source data are provided as a source data file.

placing the RGD on the oligonucleotide with the surface anchoring group (Fig. 1d). We then created a library of these force non-responsive DMC by shifting the internal RGD modification to different points (sites 1–5 in Fig. 1d) and tested them in silico by applying forces using oxDNA. We kept the anchor point constant but varied the RGD pulling position along the TD. As expected, as the stress points were separated, a greater fraction of base-pairs were denatured due to DNA overstretching (Fig. 1d, Supplementary Fig. 4). For example, pulling on the DMC on position 5 led to 41% of base-pair to denature, which is in contrast to pulling on position 1 with the RGD ligand 14 bp from the anchor terminus, which showed only 14% of base pairs denatured (Fig. 1f, Supplementary Video 4). Accordingly, we used the DMC with pulling point 1 as our force non-responsive DMC (DMC$_{rigid}$) in subsequent experiments. Unlike other DMCs, which release the FB strand upon mechanical pulling, the DMC$_{rigid}$ retains the adhesion ligand and does not interrupt mechanotransduction in a cellular context.

We also calculated and plotted the average and range of DMC pore sizes under force (Fig. 1e). Each DMC presents four pores, and each pore was approximated as an incircle that captures the cross-section of a sphere that could escape through the pore. The pore radii for all DMCs averaged around 2 nm and did not exceed 2.3 nm for the four structures tested if the force remained below the rupture force. This indicates that cargo >2.3 nm would not leak as long as the oligonucleotides comprising the DMC do not rupture. Interestingly, modeling also predicts that only the DMC$_{39pN}$ undergoes significant structural deformation upon unthreading of the FB strand. This is because the FB strand is longer for the DMC$_{39pN}$ structure, and FB strand denaturation leads to loss of TD edge integrity which releases encapsulated cargo (green zone in Fig. 1e).

We reported in our past studies that integrin forces are oriented at ~30°–50° angles to the substrate plane[40]. Hence, the effect of force direction on the DMC$_{39pN}$ was explored to assess changes in rupture dynamics and force tolerances (Supplementary Fig. 5). Modeling showed that FB strand rupture resulted in an opened DMC$_{39pN}$ with a loss of 30–51% of base pairs across all tested force orientations.

Importantly, when the quencher–fluorophore distances were examined, they were separated in all cases by more than 10 nm, which would generate a positive tension signal (Supplementary Fig. 6). Taken together, modeling predicts that DMC$_{39pN}$ releases encapsulated contents upon experiencing forces >39 pN regardless of vector orientation, and thus suggests that this design is suitable for force-mediated delivery and motivated our subsequent experiments.

## Synthesis and characterization of DMCs

Agarose gels and dynamic light scattering analysis confirmed the formation of DMC$_{39pN}$ as indicated by reduced mobility and $9.3 \pm 1.9$ nm particle diameters, which is consistent with the literature (Supplementary Fig. 7)[41]. Having confirmed the formation of DMCs, we next introduced chemical modifications to enable surface tethering, presentation of the integrin ligands, and tagging with fluorophore-quencher reporters to monitor force-induced rupture (Fig. 2a).

The first challenge in engineering DMCs for force-triggered drug release pertains to generating a sufficiently high density of RGD-origami structures on the surface. Cell adhesion to a substrate requires a minimum critical inter-ligand spacing of <60 nm[11]. DNA tension sensors are typically anchored using biotin–streptavidin binding. However, to withstand the large forces generated by integrin receptors, we used covalent chemistry as the biotin-streptavidin complex dissociates upon experiencing ~20 pN forces for ~20 min[42,43]. Specifically, we used the methyltetrazine-trans cyclooctene (Tz-TCO) click reaction, which is bio-orthogonal and has a high second-order rate constant (~$10^3$ M$^{-1}$ s$^{-1}$), thus ideally suited for generating dense DMC monolayers[44]. Accordingly, we covalently linked DMCs to surfaces using an outward-facing terminal Tz group which could be rapidly "clicked" to TCO functionalized glass surfaces. Figure 2b shows fluorescence microscopy images of Cy3B-DMCs clicked to a TCO slide (20 nM, 1 h and r.t.). By using a fluorescence calibration curve, we determined the DMC density to be $2890 \pm 76$ DMC per µm$^2$ (inter-DMC spacing of ~18 nm) at 50 nM, approaching the maximum packing density given the dimensions of the DMC (Supplementary Fig. 8).

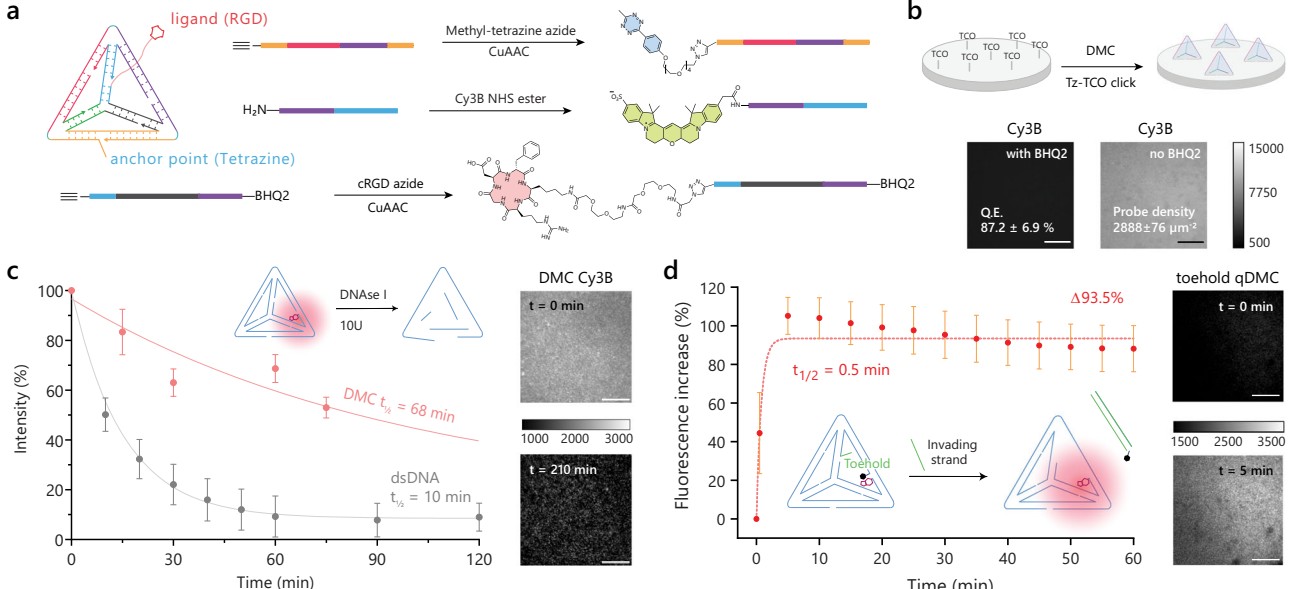

**Fig. 2 | Synthesis and functional characterization of DMCs. a** Schematic of DMC strand conjugation to methyltetrazine (Tz), Cy3B and cRGD peptide using reactive ester and copper-catalyzed azide/alkyne cycloaddition reaction (CuAAC). **b** Surface functionalization of DMCs using trans-cyclooctene (TCO) and Tz click chemistry. Images show quenched (with BHQ2) and unquenched DMCs (no BHQ2). The quenching efficiency (Q.E.) of the DMCs and the density of DMCs functionalized at 50 nM are also denoted in these images, respectively. **c** Time course of DMC (red) and dsDNA (gray) degradation using 10U DNase I and 10% FBS in DMEM. The half-life was obtained from a decay curve fit to the data. **d** Toehold-mediated dequenching of qDMCs grafted to the surface with a bulk DMC concentration of 10 nM. The calibration bars accompanying images indicate fluorescence values and display LUT. All data points are presented as mean ± SEM and represent 3 independent experiments ($n = 3$). Scale bar, 20 µm. Source data are provided as a source data file.

We also compared the fluorescence intensity of quenched and non-quenched DMCs to determine that these structures displayed a quenching efficiency of $87 \pm 7\%$. This value indicates an 8-fold enhancement in fluorescence upon DMC rupture. While the enhanced nuclease resistance of TDs is well-documented for soluble species[45,46], less is known when these nanostructures are immobilized. To test if DMCs on the surface displayed enhanced nuclease resistance compared to dsDNA, we grafted surfaces with fluorescent DMCs and dsDNA and measured loss of signal following treatment with 10 U/mL DNase I in 10% FBS. We found that DMCs had a $t_{1/2} = 68$ min, which is ~7 times as nuclease resistant as that of dsDNA with $t_{1/2} = 10$ min (Fig. 2c).

To validate that DMCs are functional on surfaces, we designed a toehold displacement reaction triggered by an invader strand leading to the dissociation of the FB strand. We quantified the yield of the displacement reaction by using fluorophore-quencher tagged DMCs labeling the FB strand (Fig. 2d). Indeed, we saw a rapid 94% increase in Cy3B fluorescence on the surface within 10 min after adding 1 μM invading strand. Thus, DMCs are responsive and functional on our surfaces.

We next designed DMCs that respond to cell-generated forces (Fig. 3a). Here, we employed a FB strand presenting a 5′ RGD at one terminus and a 3′ BHQ2 on the other end (Supplementary Fig. 2). An adjacent strand was tagged with Cy3B to form a quenched 39 pN DMC (qDMC$_{39pN}$). We generated a dense monolayer of qDMC$_{39pN}$ and then seeded NIH3T3 fibroblasts expressing GFP–Paxillin on these surfaces. Within 30 min of seeding, we observed dequenching and rupture of qDMC$_{39pN}$ under cells with a pattern that resembled the size and aspect ratio of focal adhesions (Fig. 3a). GFP–Paxillin colocalized with the DMC signal, validating the tight association between focal adhesion formation and force transmission (Fig. 3b).

To demonstrate that DMCs show cell-type specific responses, we seeded non-invasive breast cancer cells (MCF-7), non-cancer mammary cells (MCF-10A), and highly invasive breast cancer cells (MDA-

MB-231) on qDMC$_{27pN}$. MCF-7 and MDA-MB-231 showed greater cell spreading area on qDMC$_{27pN}$ surfaces compared to MCF-10A (Fig. 3c, d). In contrast, MCF-10A and MDA-MB-231 showed threefold and sixfold greater force-induced DMC rupture compared to that of the MCF-7 cells on DMC$_{27pN}$ (Fig. 3c, e). This result is consistent with past studies that showed that MCF-7 and MDA-MB-231 have greater traction forces compared to MCF-10A[47,48]. MCF-10A and MDA-MB-231 display $\alpha_v\beta_3$ integrins while MCF-7 lacks the receptor[49–51]. Thus, the observed differences in traction forces are likely attributed to the differential expression levels of $\alpha_v\beta_3$ integrins, which have 2 nM affinity towards the cRGDfK ligand[52]. Taken together, DMCs show cell-specific responses leveraging the chemical specificity of the ligand coupled with the force transmission through that ligand–receptor complex.

## Cargo encapsulation and in vitro evaluation of DMCs

The TD structure has been widely used for drug delivery because its nanoscale size minimizes clearance and displays enhanced cell uptake, while the constrained geometry offers improved nuclease resistance[53]. As a proof-of-concept for encapsulation, we used fluorescently labeled 10 kDa dextrans (radius ~2.3 nm[54]), as TD can hold cargoes of up to ~2.6 nm radius (Supplementary Fig. 9A, B)[33]. Encapsulation of dextran was achieved by annealing the DMC oligonucleotide strands with the dextran cargo and removing the excess using a 30 kDa size-exclusion cutoff filter given the 60 kDa of the cargo. DMCs annealed with varying concentrations of dextrans had up to 44% encapsulation efficiency (Supplementary Fig. 9D). Size exclusion chromatography of the Dex$_{Cy5}$ loaded DMC, DMC[Dex$_{Cy5}$], showed 260 nm and 646 nm peaks with the same retention time as an empty DMC, providing further evidence for cargo encapsulation (Fig. 4a, Supplementary Fig. 9E, F).

We next validated that cargo-loaded DMCs were functional by loading a BHQ2 quencher-modified DMC with Dex$_{Cy3B}$ (DMC[Dex$_{Cy3B}$]). Upon adding an invading strand (1 μM) complementary to the BHQ2 oligo, we noted a rapid (<10 min) increase in

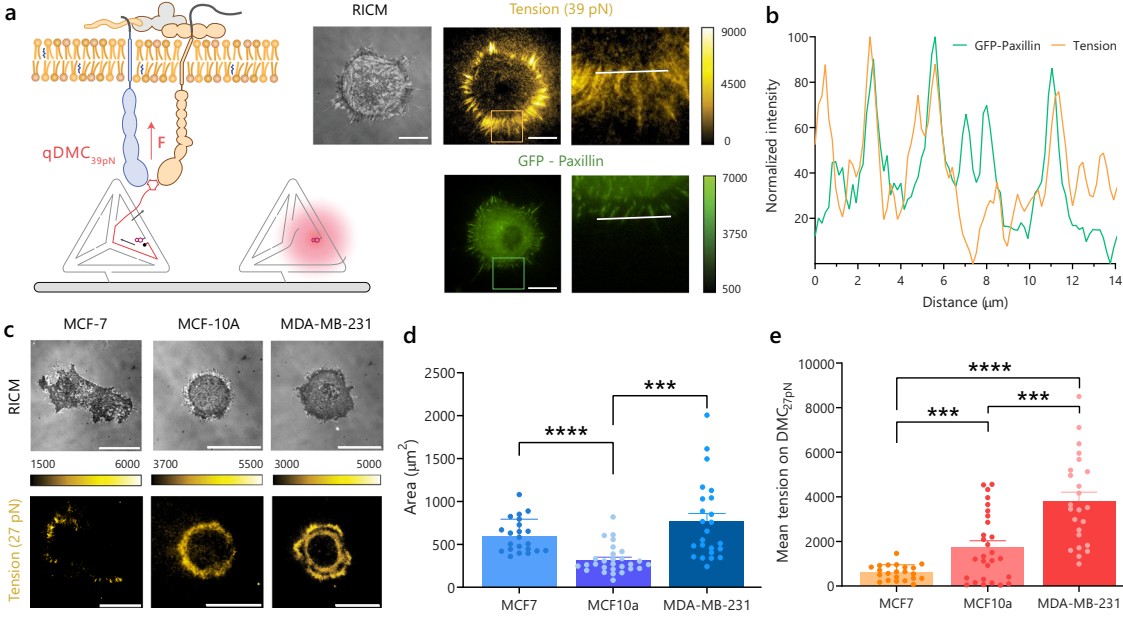

**Fig. 3 | Receptor force-induced mechanical rupture of DMCs. a** Schematic for force-mediated dequenching of DMCs due to integrin forces and microscopy images of NIH3T3 cells expressing GFP-Paxillin seeded on qDMC$_{39pN}$ surfaces in 0.1% FBS after 30 min. RICM (reflection interference contrast microscopy) channel was used to observe spreading and image analysis (n = 3 biological replicates). **b** Line scan of normalized intensities from Cy3B (tension) and GFP (GFP–Paxillin) channels from the region indicated in (**a**). **c** Microscopy images, **d** cell spreading area (***$P = 0.0003$, ****$P = 0.00001$), **e** mean tension per cell of MCF-7, MCF-10A,

and MDA-MB-231 cells (**$P = 0.0017$, ***$P = 0.0005$, ****$P = 0.0000002$) in 1% FBS DMEM on qDMC$_{27pN}$ after seeding on the surface for 1 h (n = 22, 28, and 24 cells observed respectively over 3 biological replicates, Brown–Forsythe and Welch ANOVA test, adjusted for multiple comparisons). Fluorescence indicates DMC rupture due to integrin receptor forces. All graphs are presented as mean ± SEM. Scale bar, 20 μm. The calibration bars accompanying images indicate fluorescence values and display LUT. Source data are provided as a source data file.

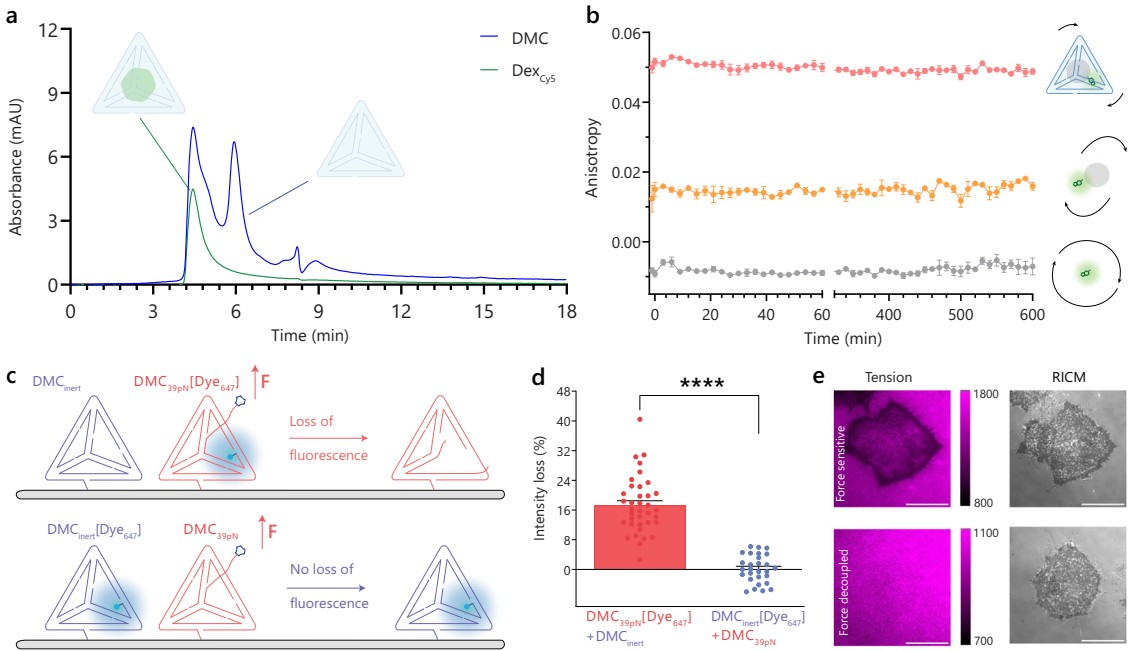

**Fig. 4 | DMC encapsulation of cargo and force-sensitive rupture under cells.**
**a** Size exclusion chromatograph of DMC[Dex$_{Cy5}$] where blue and green indicate absorbance at $\lambda = 260$ nm, and $\lambda = 646$ nm, respectively. **b** Fluorescence anisotropy of DMC[Dex$_{488}$] (red line), free Dex$_{488}$ (orange line), and free Alexa488 dye (gray line) over 10 h ($n = 3$ independent samples). **c** Schematic of a binary mixture of DMC$_{39pN}$ [Dye$_{647}$] + DMC$_{inert}$ showing the mechanism of fluorescence loss due to integrin forces. **d** Plot quantifying per cell loss of fluorescence normalized to

background under HeLa cells in DMEM (0% FBS) after 1 h ($n = 37$ and 28 cells examined over 3 biological replicates, ****$P < 0.0000001$, two-tailed Student's $t$-test). **e** Representative images of HeLa cells quantified in (**d**). All graphs are plotted as mean ± SEM. Scale bar, 20 μm. The calibration bars accompanying images indicate fluorescence values and display LUT. Source data are provided as a source data file.

fluorescence, confirming the toehold exchange proceeded and the Dex$_{Cy3B}$ was dequenched (Supplementary Fig. 9C). Our data confirms Dex$_{Cy3B}$ encapsulation and the rapid functional response of DMCs.

To further confirm encapsulation and test stability we used fluorescence anisotropy which can distinguish between encapsulated and free cargo. Fluorescence anisotropy showed distinct values that identified free Alexa488 ($r = -0.08$), Alexa488 conjugated dextran (Dex$_{488}$; $r = 0.015$), and Dex$_{488}$ encapsulated in DMC (DMC[Dex$_{488}$]; $r = 0.049$) (Fig. 4b). Importantly, we observed no change in anisotropy over 10 hours indicating the exceptional stability of cargo-encapsulated DMCs (Fig. 4b). Taken together, our data shows that DMCs can retain cargoes with minimal leakage and are thus appropriate for force-activated delivery applications.

To further validate that the DMCs are ruptured exclusively due to mechanics, we prepared four types of FB strands and four corresponding DMCs: one that incorporated RGD and Alexa647 (DMC$_{39pN}$ [Dye$_{647}$]), a second with RGD (DMC$_{39pN}$), a third that lacked RGD but had the dye (DMC$_{inert}$ [Dye$_{647}$]), and finally a DMC$_{inert}$ that was unlabeled (Fig. 4c). Cells grown on the binary mixture of DMC$_{39pN}$ [Dye$_{647}$] + DMC$_{inert}$ showed 17% loss of Alexa647 fluorescence after 1 h of seeding HeLa cells. In contrast, cells grown on DMC$_{inert}$ [Dye$_{647}$] + DMC$_{39pN}$ did not show a change in fluorescence (Fig. 4d, e). These data confirm that DMCs respond to cellular forces by mechanical denaturation and suggest the feasibility of releasing encapsulated cargo.

To emulate the delivery of macromolecular cargo to cells exerting high integrin forces, we grafted surfaces with DMCs encapsulating Atto647N dextrans (DMC[Dex$_{647N}$]). To distinguish between non-specific DMC degradation and force-mediated release, we used a mixture of DMCs as follows: (1) DMC$_{39pN}$ [Dex$_{647N}$] + DMC$_{inert}$ to quantify force-mediated delivery (Fig. 5a); (2) DMC$_{inert}$ [Dex$_{647N}$] + DMC$_{39pN}$ to measure non-specific cargo delivery (Fig. 5b); and (3) DMC$_{rigid}$ [Dex$_{647N}$] + DMC$_{inert}$ to measure uptake when DMC is

clamped closed (Fig. 5c). The choice of binary mixtures of DMCs ensured that the surfaces have identical RGD and DMC density thus offering a chemically identical surface for anchoring RGD groups while displaying a differential response to force.

HeLa cells were seeded to these surfaces having 10% FBS in DMEM and the uptake of dextran was measured by running flow cytometry after 2 h. We performed the force-induced uptake measurement at the 2 h time point because prior work showed DNA tension signal 30–90 min after cell seeding on a substrate[19,55]. We observed that the force-mediated uptake of Dex$_{647N}$ (group 1) was ~2-fold or greater than that of the controls (groups 2 and 3) (Fig. 5d, e). The higher uptake observed in group 2 compared to 3 may be due to the documented enhancement in cell migration on low force-threshold surfaces, which would lead to greater DMC encounter by cells and increased uptake[56]. We also wanted to evaluate the collateral uptake into cells with low force among a high-force cell population. To measure collateral uptake, we co-cultured MEF cells expressing GFP-tagged vinculin (Vin-GFP) and MEF cells with vinculin knocked out (VinKO) on a surface functionalized with DMC$_{39pN}$ [Dex$_{647N}$]. We observed that MEF cells that express vinculin (high force phenotype) had 2-fold higher uptake compared to MEF cells without vinculin (low force phenotype) (Fig. 5f, g, Supplementary Fig. 10). Taken together, the data demonstrates that the uptake was predominantly due to cell receptor forces and specific to cell population with receptor forces higher than the DMC force threshold.

**DNA mechanocapsules for force-responsive RNA knockdown**
We then aimed to demonstrate mechanically-triggered delivery of nucleic acid therapeutics to cells. In principle, this approach could be used to target specific cells that display high traction forces, which is often associated with invasive cancer cells[47,48]. Mechanically-mediated delivery could augment conventional targeting strategies that employ cancer biomarkers such as folate receptors[57] and HER2[58] receptors.

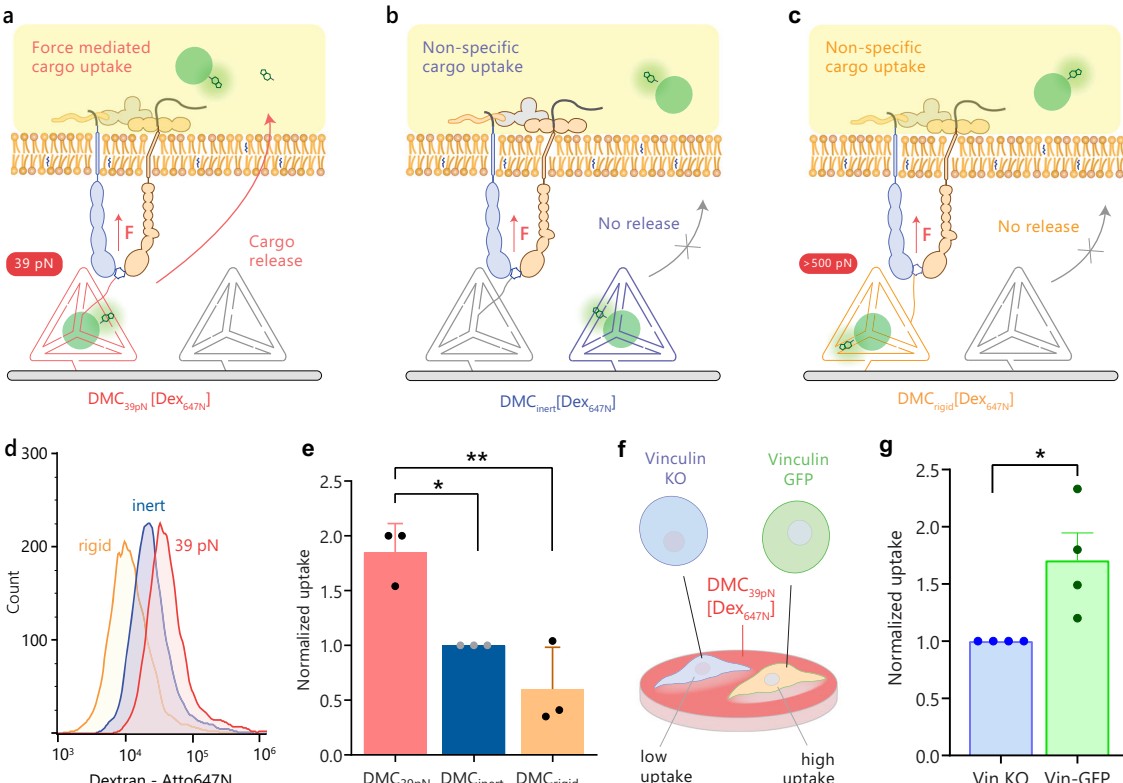

**Fig. 5 | Force-mediated delivery to cells using DMCs. a–c** Schematics showing three binary surfaces used to validate force-induced uptake of dextran cargo. **d** Representative flow histograms of HeLa cell uptake of $Dex_{647N}$ in 10% FBS DMEM media after 2 h. **e** Normalized flow data from (**d**). The plot shows median fluorescence intensity (MFI) for HeLa cells on $DMC_{39pN}$ [$Dex_{647N}$] + $DMC_{inert}$ (red), $DMC_{inert}$ [$Dex_{647N}$] + $DMC_{39pN}$ (blue), and $DMC_{rigid}$ [$Dex_{647N}$] + $DMC_{inert}$ (orange) surfaces. Data normalized to the $DMC_{inert}$ group ($n = 3$ biologically independent experiments, **$P = 0.0069$, *$P = 0.0269$, one-way ANOVA, adjusted for multiple comparisons). **f** Schematic of a co-culture of VinKO and Vin-GFP MEF cells on the same $DMC_{39pN}$ [$Dex_{647N}$] surface. **g** Normalized uptake from the VinKO and Vin-GFP MEF cell co-culture. Data normalized to the VinKO group ($n = 4$ biologically independent experiments, *$P = 0.0267$, two-tailed Student's *t*-test). Each data point represents Median ± SEM from a single flow cytometry experiment. Source data are provided as a source data file.

Here we employed an antisense oligonucleotide (ASO) drug that inhibits the mRNA encoding for hypoxia-inducible factor 1α (HIF-1α), which is a transcription factor that activates an array of genes under hypoxic conditions[59]. Importantly, HIF-1α is a key gene that aids in cancer cell survival, and HIF-1α ASOs were in clinical trials to treat solid tumors[36] and also for hepatocarcinoma[60,61]. Hence, we conjugated this ASO to the FB strand of $DMC_{39pN}$ to construct the HIF-1α interwoven $DMC_{39pN}$ [HIF-1α] and $DMC_{39pN}$ [(HIF-1α)$_2$] (Fig. 6a) which have close to 100% drug encapsulation efficiencies due to covalent conjugation. The $DMC_{39pN}$ [HIF-1α] and $DMC_{39pN}$ [(HIF-1α)$_2$] liberate one and two ASOs, respectively. The release occurs in a force-induced manner which will likely be endocytosed by integrin recycling pathways[62,63] as the ASOs are covalently linked with the integrin-binding RGD peptide (Fig. 6b). To control for non-specific release, DMCs were designed with ASOs linked to a strand adjacent to FB strand ($DMC_{inert}$ [HIF-1α]) which would not release upon force application (Fig. 6c). To maintain identical surface densities of DNA and RGD the DMCs were formulated as a binary mixture: (1) $DMC_{39pN}$ [(HIF-1α)$_2$] + $DMC_{inert}$ (2) $DMC_{39pN}$ [HIF-1α] + $DMC_{inert}$ (3) $DMC_{inert}$ [HIF-1α] + $DMC_{39pN}$. HeLa cells were cultured on these DMC-grafted surfaces with an initial 6-hour serum starvation phase followed by 10% serum addition and growth. The mRNA levels of these cells were then quantified using RT-qPCR after 24 h. The force-responsive $DMC_{39pN}$ [(HIF-1α)$_2$] and $DMC_{39pN}$ [HIF-1α] produced a 37 ± 8% and 20 ± 7% knockdown of HIF-1α RNA levels, respectively, whereas the unresponsive $DMC_{inert}$ [HIF-1α] had almost no change in mRNA levels (-5% reduction) compared to cells cultured on a $DMC_{39pN}$ (Fig. 6d). Taken together these results demonstrate the capabilities of DMCs in releasing cargo to diseased cells specifically to

those displaying high receptor force phenotypes. Importantly, these experiments represent a proof-of-concept demonstration, and virtually any validated ASO or approved drug could be delivered using DMCs in a mechanically selective manner.

## Discussion

We have created a DNA mechanocapsule (DMC) that first requires chemical recognition and binding between a cell surface receptor and ligand, which is then followed by transmission of a specific magnitude of force to release drug cargo. The DMC nanostructure platform is modular and can be programmed to rupture at tunable force magnitudes in the Piconewton regime to release a wide variety of cargoes. The strength of our approach rests on drug activation using mechanical cues that are innately generated by cells as opposed to the precedent of using physical cues such as ultrasound, thermal, optical, and magnetic actuation, which are not unique markers of the disease state.

Our modeling showed that by simply shifting the adhesion ligand to various points on the DMC, the force threshold for cargo release could be finetuned between 27 and 44 pN to target specific cells and their mechanotransduction pathways. Given that the molecular loading rates are currently unknown and the rupture probability is loading rate dependent, these values provide a metric to compare the mechanical stability of different structures. Anisotropy and fluorescence microscopy measurements confirm that DMCs are robust delivery vehicles with high stability and triggerable cargo release on demand. We demonstrated force-specific delivery of fluorescently labeled dextran as a macromolecular model cargo to cells. Dextran was used as a scaffold to load fluorophores in DMC, and in principle, a wide

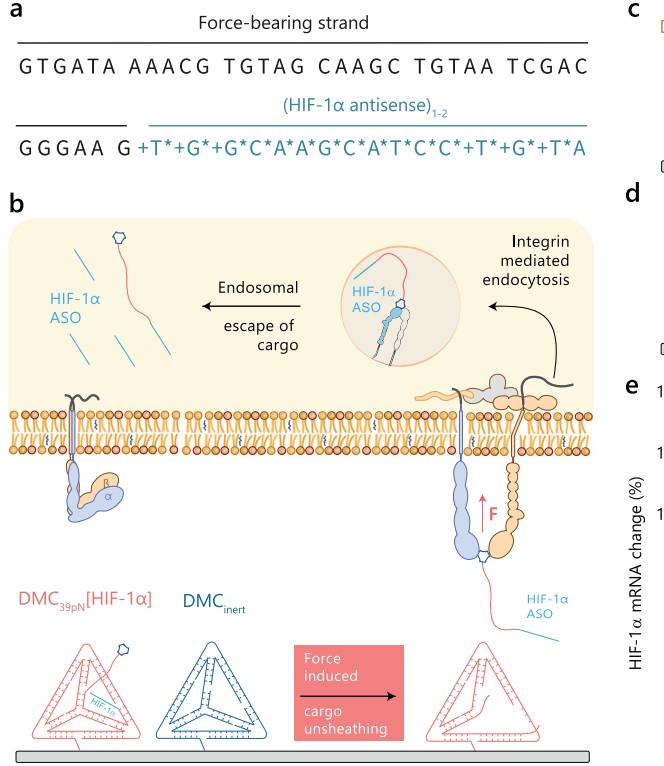

**Fig. 6 | Force-triggered ASO delivery. a** DNA sequence of RGD conjugated force-bearing strand linked to the HIF-1α ASO (blue) with LNA (+) and PS modifications (*). **b** Schematic of force-induced unsheathing of HIF-1α ASO from DMC$_{39pN}$ [HIF-1α] which can potentially be taken up into cells by endocytosis of the integrin-cRGD-HIF-1α ASO complex. **c**, **d** Schematics of force-induced unsheathing of two HIF-1α ASOs from DMC$_{39pN}$ [(HIF-1α)$_2$] and control experiment with DMC$_{39pN}$ where release of HIFα ASO is placed in the DMC$_{inert}$ [HIF-1α] and is decoupled from the force-induced rupture. **e** Plot of HIF-1α mRNA levels quantified with RT-qPCR. The data is plotted as percent change compared to cells cultured on DMC$_{39pN}$ under similar conditions after 24 h ($n = 6$, 8, 10, and 14 independent samples measured across 3–5 biological replicates, one way ANOVA, individual $P$ values are reported without adjustment ****$P = 0.000006$, **$P = 0.0034$, *$P = 0.0310$, ns–not significant). HIF-1α mRNA levels were measured relative to 18S mRNA. Source data are provided as a source data file.

variety of small molecules conjugated to dextran can be encapsulated using the same strategy. We show that DMCs have chemo-mechanical selectivity and using a small pilot of three cell lines, show that DMCs can discriminate cell types based on their mechanical phenotype. We also demonstrated a complementary strategy of creating covalently encapsulated drug molecules that can be released from the DMC in response to mechanical inputs, which leads to a 7.4-fold differential between force-responsive and inert DMCs in triggering knockdown of mRNA.

Another key advantage of mechanically mediated delivery is that the total concentration of the drug is massively reduced. For example, we estimate that only femtomoles of the ASO drug are immobilized on the chip surface, and generating this magnitude of knockdown from DMC$_{39pN}$ [HIF-1α] and DMC$_{39pN}$ [(HIF-1α)$_2$] would require bulk ASO concentration of ~50 nM and ~100 nM, respectively. This represents a 1000-fold reduction in the amount of required drug because the delivery is highly localized (Supplementary Fig. 11). It is important to state that in order to use our force-induced drug delivery strategy in vivo, DMCs must be tethered to the ECM. There are existing strategies to target peptide and antibody drugs to the ECM[64–67] and these approaches suggest feasibility (Supplementary Note 1). Taken together, our results demonstrate that DMCs can be used to deliver therapeutic cargo in a force-selective manner to target cells with specific biophysical phenotypes.

## Methods
All Oligonucleotides and primers for RT-qPCR were custom-synthesized (Tables S2 and S3) by Integrated DNA Technologies

(Coralville, IA, USA). The cRGDfK peptide (PCI-3696-PI) was purchased from Vivitide (MA, USA). DMCs were characterized using DLS on a NanoPlus DLS Nano Particle Size (Micromeritics Instrument Corporation, Norcross, GA, USA) instrument. Agarose gels were visualized on the Amersham Typhoon laser scanner (Cytiva, Marlborough, MA, USA). Gels were run on a Bio-Rad Powerpac Basic Electrophoresis Supply (Hercules, CA, USA). P2-gels (1504118) were acquired from Bio-Rad (Hercules, CA, USA). TCO-NHS, TCO-PEG$_4$-NHS, Methyltetrazine-PEG$_4$-azide, Tris- hydroxypropyl triazolyl methylamine (THPTA, 1010-100), and Methyltetrazine-PEG$_4$-NHS ester were obtained from Click chemistry tools (Scottsdale, AZ). Azido acetic NHS ester was obtained from BroadPharm (San Diego, CA). ATTO chambers, SMCC (22360), 6× Loading dye (R0611), Bond-Breaker TCEP Solution (77720), and 10 kDa amino dextran (D1860) were obtained from ThermoFisher Scientific (Waltham, MA, USA). Amicon Ultra-0.5 mL Centrifugal Filters 30 kDa (UFC503096) and Atto 647N NHS ester (18373-1MG-F) were purchased from Millipore Sigma. N,N-dimethylformamide (DMF, 227056), N,N-diisopropylethylamine (DIPEA, 496219), (3-Aminopropyl)triethoxysilane (APTES, 440140, 99% purity), sodium L-ascorbate (A4034-100G), 1-Methyl-2-pyrrolidinone (328634), were purchased from Sigma-Aldrich (St. Louis, MO, USA). Cy3B-NHS ester (PA63101) was purchased from GE Healthcare Life Sciences. Fetal Bovine Serum (35-015-CV), Penicillin–Streptomycin (30-002-CI), Trypsin EDTA (25-053-Cl), DMEM (45000-336) from Corning (Tewksbury, MA, USA). Human recombinant Insulin and MEGM Mammary Epithelial Cell Growth Medium BulletKit was purchased from Lonza (USA). Cholera Toxin - azide free (19654) was purchased from Cayman Chemicals.

Triethylamine Acetate (TEAA, 2.0 M) solution was purchased from Glen Research. Flow cytometry was performed on Beckman Coulter Cytoflex (Pasadena, CA, USA). PerfeCTa SYBR Green FastMix, ROX (95073-05K) was purchased from VWR. RNeasy Mini Kit (74106) was purchased from Qiagen. qPCR was performed with the following: RNeasy Mini Kit from Qiagen (74106; Hilden, NRW, Germany), High-Capacity cDNA Reverse Transcription Kit from Applied Biosystems (4368814; Foster City, CA, USA), and PerfeCTa SYBR Green FastMix Reaction Mix from QuantaBio (101414-278 [VWR]; Beverly, MA, USA).

### DMC annealing

All the strands of the DMC were prepared at a concentration of 200 nM and mixed with 100 μL of 10× TM buffer, which was then diluted to 1 ml. The DMCs were annealed at 10–100 μL volumes in a thermal cycler using the following annealing protocol: heat to 95 °C for 2 min; hold at 95 °C for 5 min; cool down to 4 °C over 10 min and then hold at 4 °C until it was added to the TCO functionalized surfaces. For dextran encapsulation, the DMCs were annealed with an excess of dye-labeled 10 kDa dextran (typically 10 μM) using the following protocol: heat to 95 °C for 2 min; hold at 95 °C for 5 min; rapidly cooled down to 4 °C and then held at 4 °C.

### Agarose gel for DMC formation

In total, 1.75 g of agarose was dissolved in a conical flask with 1× TAE (50 ml) and was microwaved until its completely dissolved to produce a clear solution. It was allowed to cool for a few minutes and then 5 μL of SYBR gold dye was added. It was then cast with a 15-well comb in the dark at r.t. for 2 h. The wells were loaded with 6 μL of the sample (1 μL of 6× loading dye + 5 μL DNA sample) and were run at 100 V for 1 h and visualized using SYBR gold stain.

### Dynamic light scattering of DMCs

In total, 1 mL of DMC (1 μM) was annealed and transferred to a 1 ml cuvette. The cuvette center was determined using a cell center detector function in the instrument. Dynamic light scattering was recorded in a cuvette with a 1 cm path length with a 50 μm pinhole with 10 accumulations at a 165° scattering angle at 25 °C.

### Trans-cyclooctene-functionalization of glass slides

Glass slides were washed with water and then fried using freshly prepared piranha solution (30% v/v $H_2O_2$ [9.8 M] in conc. $H_2SO_4$) for 30 min. The slides were washed with 18.2 MΩ water (×3) and then with ethanol (×3). The washed slides were immersed in 1% v/v (3-Aminopropyl) triethoxysilane in ethanol and stirred for 30 min. Slides were then washed with ethanol (×4) and then baked in a hot air oven for 60 min. TCO-NHS (trans cyclooctene) or TCO-PEG$_4$-NHS ester in DMSO (-10 mM) was added to one glass slide (50 μL TCO soln. for 25 mm circular slides; 200 μL for 25 × 75 mm slides) and sandwiched with another glass slide on top and allowed to react for 12 hrs. Finally, the slides were washed with ethanol (×2) and mounted on ATTO chambers for use.

### DMC functionalization on TCO-glass slides

DMCs functionalized with a methyl-tetrazine on the anchoring strand were annealed at 200 μM. TCO-functionalized glass circular coverslips were mounted on ATTO chambers after washing with ethanol. The mounted coverslips were then washed with a 10 ml continuous flow of 18.2 MΩ water, followed by 10 ml of 1× TM buffer. In total, 50 μL of annealed DMCs were added to the 1× TM on TCO surfaces (final DMC concentration -16 nM) and allowed to click in the dark at room temperature for over 60–90 min.

### Cell culture

NIH3T3 and NIH3T3 cells transfected with GFP–Paxillin were cultured in DMEM (10% CCS, 1% Penicillin–Streptomycin) at 37 °C in an incubator with a humidified 5% $CO_2$ atmosphere. MEF, HeLa, and MDA-MB-231 cells were cultures under similar conditions but with 10% FBS DMEM and 1% Penicillin–Streptomycin. MCF-10A cells were cultured using Mammary Epithelial Cell Growth Medium supplemented with 100 ng/ml cholera toxin and BulletKit from Lonza. MCF-7 cells were grown in 10% FBS DMEM and 1% Penicillin–Streptomycin added with 0.01 mg ml$^{-1}$ human recombinant insulin. Cells were cultured as per ATCC specifications and passaged either using 0.25% trypsin (5 min) or 50 mM EDTA in 1× PBS (10 min) and reseeded in a new flask with suitable media at lower density.

### Coarse grain oxDNA simulations

The simulations were modeled using the oxDNA2 model (version 2.4 published in June 2019). We ran the MD simulations on CPUs and GPUs. The following parameters were used extensively in the simulations run on GPUs.

```
backend = CUDA                    interaction_type = DNA2
backend_precision = mixed         use_average_seq = 1
CUDA_list = verlet                verlet_skin = 0.05
CUDA_sort_every = 0               salt_concentration = 0.156
use_edge = 1                      thermostat = john
edge_n_forces = 1                 newtonian_steps = 103
sim_type = MD                     diff_coeff = 2.5
T = 37C                           dt = 0.005
steps = 1e9
time_scale = linear
```

**Force–extension curves**. The DNA sequences of the DMCs were imported into oxDNA format. The DMCs were then minimized and relaxed using the input parameters from published literature[5] and examples available at dna.physics.ox.ac.uk website. The DMC rupture was modeled by adding harmonic traps (which acts like springs) to the cRGD and the methyltetrazine attachment points on the DMCs. Each trap was assigned a stiffness constant of 11.42 pN/nm, and the trap attached to the cRGD was moved at a specified loading rate with respect to the other fixed trap. The effective stiffness constant of the two traps in series can be calculated using:

$$\frac{1}{k_{eff}} = \frac{1}{k_1} + \frac{1}{k_2} \tag{1}$$

where $k_1$ and $k_2$ are the stiffness constants of the two traps and $k_{eff}$ is the effective stiffness constant. The $k_{eff}$ of the system is calculated to be 5.71 pN nm$^{-1}$. The traps are moved at a rate of $5 \times 10^{-8}$ (length per unit of time in oxDNA units). This rate can be converted into SI units as shown here:

$$\text{Loading rate} = \frac{5 \times 10^{-8} \times 0.8518 \text{ nm}}{3.03 \times 10^{-12} \text{ s}} = 1.4 \times 10^4 \text{ nm s}^{-1} \tag{2}$$

The net force exerted at a given point in time was calculated by multiplying the total displacement of both the harmonic traps from respective nucleotides with $k_{eff}$. The obtained force was then projected along the axis in which the traps were moved to get the force due to harmonic traps. This force was then plotted along the trap extensions at that given time, along with an exponential moving average (EMA) of the force. Other parameters such as total number of base pairs, total energy, and specific particle distances, were also extracted for analysis. The output was analyzed using Python with the NumPy[6], SciPy[7], Pandas[8], and Matplotlib[9] packages.

**DMC pore size**. The four faces of a DMC were assumed to be triangles with the hinge nucleotides at the vertices. The macromolecular cargo inside the DMC was assumed to be spherical. The radius ($r$) of the

largest circle that can fit inside a triangle is given by the incircle of triangle formula:

$$r = 2 \times \frac{\text{Area of triangle}}{\text{Perimeter of triangle}} \qquad (3)$$

DMCs are considered leaky when the incircle radius is larger than the radius of the cargo. The positions of the hinge nucleotides were extracted from the force-extension simulation trajectories of the DMCs. The hinge nucleotide positions were used to calculate the area and perimeter of each face of the tetrahedron, which was then converted into incircle radii. The incircle radii from all four faces were then plotted as the maximum with the shaded region depicting the range between maximum and mean.

**DMC rupture dynamics.** The cRGD attachment points on the DMCs were subjected to forces along different directions. The number of hydrogen-bonded base pairs in the DMC as well as fluorophore-quencher distances, were obtained from the simulation. Various particle positions on the force-bearing strand and the anchoring strand were also extracted. The separation distance of these strands from their complementary edges on the DMC was then estimated. It was plotted as a function of time to depict DMC rupture dynamics.

## Purification of DMCs using M.W. cut-off filters
DMCs (20–100 μL) were annealed with excess dextrans and diluted to 500 μL in 1× TM buffer. Then it was transferred to a 30 kDa M.W. cut-off filter and centrifuged at 14,000×$g$ for 5 min. The filters were then added with 500 μL of 10× TM buffer and centrifuged (×3). Then the filters were added with ~200 μL of 1× TM buffer, inverted, and centrifuged at 200×$g$. The purified DMCs were then used for further experiments.

## Fluorescence anisotropy
DMCs (100 nM) were each annealed with excess Atto488-labeled 10 kDa dextran (100 μM) and filtered using a 30 kDa filter. Atto488-labeled 10 kDa dextran was purified using 3 kDa filters to remove free Atto488 dye. The purified products were subjected to a second round of M.W. cut-off filtration to ensure high purity. The concentrations of the free dye, labeled dextran, DMC[Dex$_{488}$], were all diluted to be 200 μL at 12.5 nM. To correct for background anisotropy, 1× PBS was used as blank. The fluorescence anisotropy was recorded using a Synergy H1 Biotek Plate Reader (VT, USA) at r.t. over 10 h. BioTek Gen 5 software (v3.13.15) was used for fluorescence data collection.

## DNase I degradation
DMCs with Cy3B and dsDNA (Cy3B_TD1-S3b and Tz_TD1-S4) with Cy3B were clicked to a TCO surface for 1 h. Surfaces were washed with 10% FBS DMEM and then added with 1 U of DNase I to 1 ml of DMEM media in the chambers. The surface was imaged over 24 h to visualize the kinetics of dsDNA and DMC degradation.

## Toehold strand displacement
**Opening DMCs on surface.** DMCs with a toehold and BHQ2 were clicked to a TCO surface for 1 h at a concentration of 10 nM. The surfaces were washed with 10% FBS DMEM and added with the invading strand (final conc. 1 μM) to commence the toehold-mediated opening of the immobilized DMCs. Images of the surface were acquired every 5 min over 2 h to visualize the kinetics of DMC opening.

**Rupturing DMCs in solution.** DMCs with toehold and BHQ2 were annealed with 100 μM of Cy3B labeled dextrans. The DMCs were purified twice using M.W. cut-off filters and then challenged with an invader strand (final conc. 1 μM) in solution and the change in fluorescence was read using a plate reader.

## Size exclusion chromatography
DMCs (200 nM) were mixed with Cy5 labeled dextrans (40 μM) and annealed to encapsulate the dextrans. The DMC[Dex$_{Cy5}$] was then purified, as mentioned above, using 150 mM phosphate buffer. The purified DMCs were then injected into a size exclusion HPLC column (SEC-130) with 150 mM phosphate buffer at 0.3 ml/min as an eluent.

## Fluorescence microscopy
**Imaging NIH3T3 cell tension.** Imaging was conducted with a Nikon Ti2-E microscope. The NIH3T3 fibroblast cells expressing GFP–Paxillin were seeded onto a qDMC (cRGD, BHQ2, Cy3B, and tetrazine modifications) grafted coverslip in DMEM (1% CCS, 1% Penicillin–Streptomycin) and allowed to attach to the surfaces for 30 min at 37 °C in the incubator with a 5% CO$_2$ atmosphere. The cells were then imaged at room temperature (~15 min) in RICM, FITC, and TRITC channels. The images were quantified using ImageJ2 software as described below.

**Imaging non-force mediated degradation.** DMCs (50 μL) each were annealed separately, mixed, and clicked directly to 550 μL of 1× TM TCO surfaces. After 1.5 h, the chambers were washed with 10 ml of DMEM (with 1% P/S, 0% FBS, and no phenol red). HeLa cells (5.0 × 10$^4$) were seeded to DMC grafter coverslips and allowed to spread for 1 h. It was then imaged at room temperature (~5 min) in RICM and Cy5 channels. The images were quantified using ImageJ2 software as described below. The intensity loss below each cell was normalized to its background.

**Imaging MCF-7, MCF-10a, and MDA-MB-231.** qDMC$_{27pN}$ were (50 μL) annealed separately and clicked directly to 550 μL of 1× TM TCO surfaces. After ~1.5 h, the ATTO chambers were washed with 1% FBS DMEM media with Penicillin–Streptomycin. The respective cells (5.0 ×10$^4$) were added to separate chambers and allowed to attach to the surfaces for 30 min at 37 °C in the incubator with a 5% CO$_2$ atmosphere. The cells were imaged at room temperature after 1 h in RICM and TRITC channels. The images were quantified using ImageJ2 software as described below.

**Image analysis.** NIS_elements (vS.2.1) was used for microscopy image and video recording. Fluorescence images were processed using the ImageJ2 software. The fluorescence background around a cell was averaged and subtracted from the whole image. Cell spreading was measured by the total area of the cell as observed in RICM. Fluorescence from tension probes or GFP–Paxillin was measured within the cell spread area. The brightness and contrast of microscopy images were adjusted for clearness. Quenching efficiency is given by the formula:

$$\text{Q.E.} = 1 - \frac{I_{\text{quenched}}}{I_{\text{unquenched}}} \times 100 \qquad (4)$$

## Flow cytometry
**Force-specific dextran uptake measurements using different DMCs.** A binary mixture of DMCs (50 μL) each was annealed and filtered using 30 kDa M.W. cut-off filters. Then they were mixed with an additional 150 μL of DMC$_{39pN}$ (200 nM) to compensate for losses from purification and to promote cell adhesion. The mixture of DMCs was allowed to click to TCO-PEG$_4$ surfaces in 1× TM for about 1 h and then washed with 10% FBS DMEM (with 1% Penicillin–Streptomycin and no phenol red). To the surfaces, HeLa cells (2.5 × 10$^4$) were seeded to DMC grafted coverslips and allowed to attach for 2 hr. The cells were then detached from the surfaces using 0.25% trypsin and centrifuged along with the growth media in the chambers at 400×$g$ for 3 min at 4 °C. The cells were washed (×3) in DPBS (without Ca$^{2+}$ and Mg$^{2+}$) and were resuspended in DPBS. The cells were analyzed using a flow

cytometer for to measure the uptake of Atto647N labeled dextrans. CytExpert software (v2.3) was used for flow data collection. The data were analyzed and histograms were prepared using FlowJo software (FlowJo LLC, USA).

**Differential cargo uptake in a co-culture of high and low-force MEF cells.** $DMC_{39pN}$ [$Dex_{647N}$] were (50 µL) each was annealed and filtered using 30 kDa M.W. cut-off filters. Then they were mixed with an additional 150 µL of $DMC_{39pN}$ (200 nM) to compensate for losses from purification and to promote cell adhesion. The mixture of DMCs was allowed to click to $TCO-PEG_4$ surfaces in 1× TM for about 1 h and then washed with 10% FBS DMEM (with 1% Penicillin−Streptomycin and no phenol red). To the surfaces, MEF cells ($2.5 \times 10^4$) were seeded to DMC grafted coverslips and allowed to attach for 2 h. The cells were then detached from the surfaces using EDTA (50 mM) and centrifuged along with the growth media in the chambers at 400×$g$ for 3 min at 4 °C. The cells were washed (×3) in DPBS (without $Ca^{2+}$ and $Mg^{2+}$) and were resuspended in DPBS. The cells were analyzed as above using a flow cytometer.

## Quantitative polymerase chain reaction (RT-qPCR)
ATTO chambers mounted with TCO surfaces were sterilized by placing them in a tissue culture hood with UV light for ~2 min. DMCs were folded and clicked by adding directly to TCO surfaces in 550 µL of 1× TM. After 1.5 h, the chambers were washed with 5 ml of DMEM (with 1% Penicillin−Streptomycin, 0% FBS). HeLa cells detached using to 50 mM EDTA in 1× PBS were seeded to DMC functionalized glass surfaces ($5.0 \times 10^4$ cells per surface) and allowed to adhere for 6 h in the incubator at 37 °C with 5% $CO_2$. After 6 h, media in chambers (~900 µL) was supplemented with 100 µL FBS and allowed the cells to grow for 18 h. After 24 h incubation, the media from the chambers were transferred to 1.5 mL tube cells. The chambers were added with 500 µL of 0.25% trypsin and incubated for 5 min. The cells were detached and spun down at 800×$g$ for 1 min at 4 °C. The cells were washed with 1 ml HBSS and resuspended in 300 µL RLT buffer. The cell lysate was immediately used for downstream analysis or stored at −80 °C. The cell lysates were added with 300 µL 70% ethanol and transferred to the Qiagen RNA Miniprep column. RNA was extracted from the lysate as per the manufacturer's instructions and transcribed using a high-capacity cDNA reverse transcription kit. HIF-1α mRNA levels were then quantified by RT-qPCR using PerfeCTa SYBR Green FastMix Reaction Mixes with a 0.25 µM concentration of custom-designed primers for HIF-1α and 18S (efficiency ~2). The relative quantification of HIF-1α mRNA levels was performed using the $\Delta C_t$ method with 18S rRNA as an internal control. LightCycler 96 Application Software (vl.1.0.1320) was used for RT-qPCR data collection and cycle value ($C_q$ value) estimation.

## Oligonucleotide modifications
**NHS-ester amine-DNA coupling.** Amine-modified DNA was dissolved in 10 µL water (1 mM) and added with 2 µL of 1 M $NaHCO_3$. NHS-modified functional group was dissolved in 10 µL DMSO (Cy3B-NHS, Atto647N-NHS, etc.) or 10 µL acetonitrile (Methyltetrazine-NHS) or 40 µL DMF/NMP (Succinimidyl-4-(N-maleimidomethyl) cyclohexane-1-carboxylate) to obtain a final concentration of 10−100 mM. These solutions were mixed and reacted at r.t. for 30−60 min. The reaction was quenched with the addition of 100 µL water and was filtered with P2 gel to remove excess NHS reagent and purified using HPLC. Integrin ligand c(RGD)fk-($PEG_2$)$_2$-$NH_2$ can be functionalized in a similar manner by reacting it with azido acetic acid NHS in DMSO for 12 h and directly purifying using HPLC.

**Thiol-SMCC conjugation protocol.** Thiol DNA strand was dissolved (5 µL) in 1× PBS at pH 6.8 to obtain a 1 mM concentration and then reduced with 100 equivalents of Tris (2-carboxyethyl) phosphine (0.5 M) at room temperature for 15−20 min. To this, 2× equivalences of maleimide-functionalized DNA in 1× PBS (10 µL) were added at r.t. to react for 3−4 h. For the di-maleimide conjugation in the case of cRGD_TD6-S3a_(HIF-1α)$_2$, 3−4× equivalences of maleimide were used. The reaction mixture was filtered using P2 gel and purified by HPLC.

**Alkyne–azide conjugation.** Alkyne functionalized DNA strand was dissolved in 5 µL water to yield a 4 mM solution. Azide reagent (10 equivalences, 20 mM) dissolved in 10 µL acetonitrile was added to this. In a separate tube, $CuSO_4$ (1 equivalence, 1 µL), Tris (3-hybroxypropyltriazolyl methyl) amine (4 equivalences, 2 µL), sodium ascorbate (2 equivalence, 1 µL), and 1 µL triethylamine were mixed. This was added to the Alkyne-azide mixture and reacted at r.t. for 60−90 min. The reaction was quenched with the addition of 100 µL EDTA solution, filtered with 0.2 µm filters and purified using HPLC.

**HPLC purification of peptides and oligonucleotides.** An advanced oligonucleotide C18 column was used for the purification of chemically modified DNA. The column was run with 0.1 M TEAA in water as solvent A and acetonitrile as solvent B. DNA was purified using the following method, 10% for 3 min followed by 10−35% solvent B gradient over 25 min at 0.5 mL/min flow rate. Tetrazine conjugated DNA was purified using a similar method but using a 10−60% solvent B over 25 min at 0.5 mL/min. Peptides were purified using a Grace C18 column which was eluted with water as solvent A and acetonitrile as B, both containing 0.05% TFA. A gradient of 10−40% solvent B over 30 min at 1 mL/min was used for peptide purification. Separated fractions were dried in a vacuum concentrator overnight. The purified oligonucleotide conjugates were reconstituted in 1× TE buffer and were stored at −30 °C. The concentration of oligonucleotides and peptides was determined using its absorbance at 260 nm and 214 nm, respectively. OpenLAB CDS chem station edition from Agilent Technologies was used for HPLC data collection.

## Supported lipid bilayer preparation
To prepare Supported Lipid Bilayer (SLBs), SUVs were primed by mixing 99.9 mol% 1,2-dioleoyl-sn-glycero-3-phosphocholine (DOPC) and 0.1 mol% Texas Red 1,2 dihexadecanoyl-sn-glycero-3-phosphoethanolamine, triethylammonium salt (TR-DHPE, T1395MP, ThermoFisher Scientific) in a round bottom flask with chloroform. Lipids were then dried for 30 min under rotary evaporation followed by an ultra-high-purity nitrogen stream to remove residual chloroform. The dried lipid film was hydrated with 18.2 MΩ water (2 mg/mL) before conducting three freeze−thaw cycles. The mixture was then passed through a 10 mL LIPEX® thermobaric extruder 10 times (Evonik Industries, Essen, Germany) using an 80 nm polycarbonate filter.

**Small unilamellar vesicles.** In this calibration, the intensity of labeled oligonucleotides and small unilamellar vesicles (SUVs) in solution are compared to determine the F factor, which relates molecular brightness of the two fluorophores. To prepare SUVs in glass bottom plates, the glass was treated with a 2 M NaOH etching solution. It was then thoroughly washed with water and 1× PBS. The ratio of the calibration curve slopes was used to determine the "F factor" for the labeled oligonucleotide and the SUV samples. The F-factor was calculated as follows:

$$F = \frac{I_{Cy3B-DNA}}{I_{TR-DHPE}} \tag{5}$$

where $I_{Cy3B-DNA}$ and $I_{TR-DHPE}$ represent the fluorescence intensities of the DNA and SUV samples, respectively. The quality of the calibration curve for DNA and SUVs was assessed by measuring the linear regression between the concentration of the known oligonucleotide (or the SUV sample) and its fluorescence intensity, as deviations can indicate nonspecific adsorption.

**Bulk probe density calculation.** Supported Lipid Bilayer allows one to convert the raw fluorescence intensity of the surface to the molecular density of fluorescent molecules[10]. In this assay, lipid membranes were used as calibrated fluorescence standards based on the known documented molecular density of phospholipids within membranes[11]. To create a fluorescence calibration curve, glass was passivated with 0.1% BSA in PBS for 20–30 min. Then, SLBs with varying fluorophore concentrations were prepared by adding mixtures of labeled and unlabeled SUVs in known stoichiometries. Excess SUVs were rinsed using 1× PBS. The intensity of the SLBs was measured using epifluorescence microscopy. Using the known lipid footprint ($0.72\,nm^2$), the generated graph was used to relate the density of fluorophores to arbitrary fluorescence units. The intensity of probes on the surface was then corrected using the F factor. Using the generated density curve, we converted the fluorescence intensity of DNA to probe density per $\mu m^2$.

## Statistics

Quantitative results for experiments, unless mentioned otherwise, were presented as mean ± SEM. All statistical analysis was performed using the GraphPad Prism software package. Statistical analyses were performed using the *t*-test or analysis of variance (ANOVA) tests. *P* values were corrected for multiple comparisons unless otherwise noted as individual *P* values. *P* values were considered significant if the tested *P* value was smaller than 0.05(*), 0.01(**), 0.001(***), or 0.0001(****).

## Reporting summary

Further information on research design is available in the Nature Portfolio Reporting Summary linked to this article.

## Data availability

The oxDNA simulation, microscopy images, flow cytometry, and other forms of data generated in this study have been deposited in the Zenodo database under accession code https://doi.org/10.5281/zenodo.10052231. Source data are provided in this paper.

## Code availability

Python scripts used for oxDNA analysis are freely available on GitHub under the link https://github.com/Arventh/DNA_MechanoCapsules.

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

## Acknowledgements

K.S. acknowledges support from NIH NIAID R01AI172452 and NIGMS R01GM131099 and 1RM1GM145394. We thank Prof. Jonathan Doye, Navoneel Sen, and Hemani Chhabra at the University of Oxford for their discussions on oxDNA simulation and analysis. We thank Prof. Yonggang Ke for his thoughtful discussions on the DMC origami designs. A.V. thanks Joseph Mancuso and Vageesha Herath for their help in characterizing oligonucleotides and their assistance in surface preparation, respectively. Figures in Supplementary Note 1 were Created with BioRender.com.

## Author contributions

A.V. and K.S. conceived and designed the research. A.V. carried out most of the experiments and analyzed the data. A.V. and R.S. designed and performed the qPCR experiments. A.R. performed SLB calibration, assisted in the characterization of DMCs with NIH3T3 cells. H.O. synthesized Cy3B-labeled RGD constructs for uptake visualization. A.V. and K.S. wrote the paper. All authors reviewed and edited the paper before submission.

## Competing interests

The authors declare no competing interests.
