## [Peer Review File · Nature Communications]

DNA mechanocapsules for programmable piconewton responsive drug deliveryREVIEWER COMMENTS

Reviewer #1 (Remarks to the Author):

The work by Velusamy et al uses previously developed DNA tetrahedral designs that are functionalized with cargo (dextran or oligonucleotide) that is delivered (uptaken by cell) in a force-sensitive manner, allowing specific delivery to cells by force-induced rupture of the tetrahedron induced by the cell surface receptors. The authors suggest promising clinical applications based on specifically targeting phenotype of cells expressing integrin adhesion receptors.

In my opinion, the work is original and of interest to the broad community of readership. The theoretical modeling results, obtained by oxDNA simulations, are supported by experiments, where authors demonstrate successful delivery of the cargo from the force-ruptured tetrahedrons. I do have several questions I would like the authors to explain about the design of the nanostructures and their computational validation. Provided they can provide satisfactory explanation, I recommend publication.

1) For testing the force-rupture in oxDNA model, is it correct that in all cases the authors anchored two different points of the structure (cRGD-attachment point and methyltetrazine attachment to the surface). If used in the bulk, these structures would be interacting with targeted cells without the methyltetrazine attachment to the surface. Do the authors consider that the simulation studies would still be representative of the way that the force would be applied during mechanical pulling of the surface receptor in that case?

2) The authors use k_{eff} spring constant to calculate the force that the structure experiences during pulling. Is there a reason to calculate k_{eff} , rather than obtain the force acting on the the first and second anchored nucleotide respectively

3) The authors studied the rupture force by moving harmonic traps at constant speed and measuring the force induced on the system. They only have 1 trajectory per each system studied, so it might not necessarily be representative and might be a very rough estimate. Additionally, it is not clear if the rupture force could be in fact lower, but since the pulling

can be too fast, the actual rupture force can be over-estimated. It would be interesting if for one system (e.g. 39pN cage), authors could try pulling and half the speed and twice the speed and see if they still obtain the same rupture force.

4) I am not sure I understand the claim that DMC_rigid can withstand forces up to 500 pN. Overstretching force for DNA duplex is around 70pN or so, so this number seems to like large overestimate. In the provided figure of stretched DMC_rigid, the ssDNA seems to be completely extended, so it is difficult to me how it can still remain part of the structure and not release the cargo?

5) For clinical application, would it be an issue that the DNA origami can get absorbed by most cells by endocytosis, and not just by the specifically targeted cells with integrin receptors on their surface?

Minor points:

6) The authors use oxDNA model to study the function of the system, yet no article that describes the model is cited. Authors should provide the citations along with explanation which version and implementation of the model they used.

7) The figure resolution is rather poor and should be increased in the final version of the manuscript.

Reviewer #2 (Remarks to the Author):

In the manuscript by Salaita et al. the authors describe a mechanoresponsive DNA nanomaterial for delivering cargo to cells via programmed degradation induced by engagement of cell surface receptors that serve to pull apart the DNA nanostructure and release its encapsulated cargo. It is a very interesting concept – and is shown to work well by their data. Its translational promise is not immediately apparent but the idea is shown to be feasible through these studies and that it could be researched further to enable its potential for therapeutic application.

Comments: The anisotropy data in Fig 3B is of how many runs? Error bars or a description of the number of trials should be included.

In the experiments depicted in figure 3f-j why was a 2 hour time point chosen? Do the ratios of fluorescence uptake expected to change with time?

Regarding the comment “ this approach could be used to target specific cells that display high traction forces which is often associated with invasive cancer cells” with regards to the motivation for mechanoresponsive delivery of RNA to cells – would this play out in vivo or would this be solely in vitro? Could the mechanism of how this might be possible in the complex environment of many cells be explained?

In figure S9, the y axis title is confusing as it says “change” (similar to figure 4e in the main paper)– usually it is a plot of total mRNA expression. Written as change it seems confusing that the 50nM sample achieved a 23% expression but a 200nM treatment had 63% expression....may just be the way it is written, it is some what hard to interpret.

For the knockdown experiments the forward and reverse primer sequences used for both the reference gene and the HIF1a target should be included in the sequences table in the supplementary information.

What is the proposed mechanism of endosomal escape of the ASO tethered to the RGD ligand? Are there concerns about the stability of the oligo once released from the DMC?

Minor points: Figure quality overall is low (for example, figure 1 and others are very blurry/low resolution).

Line 38 in the introduction’s first paragraph appears to be missing a word – it looks like something is missing grammatically.

Figure 3a, the schematic showing the encapsulated cargo (green trace) should be better aligned with the green line – looks like its pointing to the blue line.

Reviewer #3 (Remarks to the Author):

The manuscript by Arventh Velusamy et al. reports a novel method to release drug molecules encapsulated in DNA mechanocapsules (DMC) by using mechanical forces generated by cells. Considering the range of forces generated by different receptors, the authors designed and simulated different tetrahedral DMCs displaying controlled rupture forces. An extensive proof of concept study is presented to validate the functionality of

DMCs under forces expressed by integrin receptors, the encapsulation, release and uptake of drug cargos. Finally, force-responsive RNA knockdown is demonstrated in cells by using functionalized DMCs.

The main novelty and contribution of this work is the demonstration of the possibility to encapsulate and release a wide variety of cargos by mechanically forces expressed by cell receptors. This type of cell targeting opens a multitude of research lines in cell biology, biochemistry and biophysics potential application to biomedicine, immunology and pharmacology.

In general, the paper is well written, and the concepts clearly presented. However, I have two main concerns regarding the DMC characterization: only a loading rate was considered in simulations and no DMC characterization by force spectroscopy.

Therefore I recommend publication pending on Major Revision.

Details supporting these concerns and further comments follow.

1. The DMCs design is clever and the authors elegantly show the possibility to tune the rupture force by changing the RDG anchoring points and the force bearing strand lengths. The response to force orientation is also analyzed, concluding that it does not influence unfolding and DMC39pN release encapsulated cargo for forces bigger than 39 pN. This value is related to the forces exerted by integrin receptor on the RDG ligand, thus making it possible to trigger the release of an encapsulated cargo by a cell in contact with DMC. While this rationale is helpful, my concern is how the value 1.4×10^4 nm/s of the pulling rate was chosen. With the stiffness $k = 5.71$ pN/nm, given in Methods 7.1, a loading rate of 8×10^4 pN/s is obtained. Moreover, since it is known that the loading rate represents a critical variable in stiffness sensing, I would expect a force analysis as a function of the loading rate with examples taken in a wide range (e.g. $10^2 - 10^5$ pN/nm). The authors mention in the text (Conclusions) that “the molecular loading rates are currently unknown”. Probably they refer to the specific RDG – integrin interaction. However, there is literature in which the loading rate values for integrin interacting with ligand are at least predicted and discussed if not measured (e.g. fibronectin). Few examples: Andreu 2021, Nat Comm: <https://www.nature.com/articles/s41467-021-24383-3> , Amar 2020, Biochem Biophys Res Comm: <https://doi.org/10.1016/j.bbrc.2020.01.149> , Kolasangiani 2022, Cells <https://doi.org/10.3390/cells11223584>, Li 2003, Biophys J: <https://doi.org/10.1016/S0006->

3495(03)74940-6

Considering the importance of the loading rate also for the rupture force, I recommend to run new simulations and discuss the results obtained.

An explanation on the elastic constants (both 11.42 pN/nm) is missing in methods and hence it is requested.

2. Fig. 1b: does the force jump have a meaning?

Fig. 1e: as regards the plot of the pore radii in fig. 1e, the figure is not clear (not visible e.g. behavior for 44 pN).

Is the range defined by min and max? Maybe representing only the mean + std values at several instants would help to show better the pore size variation under force.

3. Comments on "Synthesis and characterization of DMCs" section

- The synthesis and deposition of DMCs on substrate are clearly presented. However, the images (fig 3 b,c,d) showing quenched and unquenched DMCs are too small and poorly explained to be helpful. For instance, the unquenched fluorescence intensity seems to vary a lot. Does this mean that the deposition is not uniform? How is the intensity measured for QE ? Averaged over the area of the image? Please explain and comment. I would also suggest that these images go to SI, with a better explanation and proper size. As regards the high DMCs density on the substrate, I wonder how this condition can be handled in vivo i.e. for cells and ECM. Please comment (more than it is already done in conclusions) in terms of applicability and density required.

• Fig. 2c Time (m)? maybe (min). Same for fig 3b

• The examples shown for NIH3T3 cells and three types of breast cancer cell are illustrating well the potential of DMC activation by cellular force, although the associated figures 2 e-i can be improved:

- the images (tension and paxilin) are too small and the correlation, if any, with RICM not explained

- the min and max scale of color bars are not indicated (i.e. 5000 counts of max ?) and hence not too relevant

- is there an explanation for the tension shown in fig2g for MDA-MB-231 (internal ring)? Is

there a traction additional to the traction at the cell edge?

- the graph in fig 2f is showing two line profiles from 2e, but the line profiles are clearly not correspondent. Moreover, colocalization is not the appropriate term for this type of images (diffraction limited) where the structure sizes are much bigger than the molecular size. Thus fig. 2f does not help illustrating colocalization and should be removed or replaced with profiles taken at a high enough resolution.

- SI Fig 7 is cut at left, in c there is no error bar for 50 nM

- Fig. 2g and h : why qDMC27pN and not 39pN? what does it mean they showed greatest response for traction forces? The spread is related to substrate stiffness. Please comment more.

- MCF-7 image is very unclear, due to a bright spot- is this spot from measurement or is it an artefact?

Main concern:

although the experiments presented in this section confirm the DMC activation by cellular forces, a methodologic step is missing after the simulation of the activation forces and before the experiments with cells: the characterization of DMC activation by force spectroscopy. In my opinion this is an important step which is required to complete DMC characterization.

4. Comments on Cargo encapsulation and in vitro evaluation of DMCs and force-responsive RNA knockdown

This is a nice proof of concept of drug encapsulation and delivery by using DMC and cellular forces.

Just as a curiosity, I wonder if a 44% encapsulation efficiency can be considered sufficient and in which conditions it can be improved. Please comment.

It is mentioned as a result that force mediated uptake is 2 fold greater than that of the controls. Is this enough and promising? Can it be enhanced? Please comment.

The images in fig. 3e are too small. Reducing a bit f, g, h some space could be used to increase the size of images which otherwise are not very useful.

The force responsive RNA knockdown: the results presented in fig. 4e are not clearly commented in the text and conclusions, please review.

The argumentation on the reduction of the total concentration of drug is not clear, and hence not convincing. I suggest to bring the info from SI to main text and make the discussion clearer.

Reviewer #4 (Remarks to the Author):

The authors describe DNA mechanocapsules (DMC) as a mechanical force-responsive drug delivery system based on DNA tetrahedrons. The DNA tetrahedrons conjugated with RGD peptides that specifically interact integrin on cancer cell surface. The position of RGD was optimized so that the RGD-labeled strand can be stretched out and eventually dehybridized out of the DNA nanostructures by the force derived from the integrin-RGD interaction. This concept could be used to specifically release model drugs encapsulated or loaded in DNA tetrahedrons depending on the integrin expression level of cells. Although the concept is novel and interesting, there are issues that should be addressed before considering the manuscript as follows.

1. Force induced uptake may not only enhance the uptake of DMC39pN into the cells with high integrin expression but also facilitates uptake of DMC39pN into the cells with low integrin expression. To address this concern, the authors need to examine whether $[\text{uptake into target cell}]/[\text{uptake into non-target cell}]$ ratio is indeed improved. Also, they have to compare the ratio, $[\text{uptake of DMC39pN into cells with high expression of integrin}]/[\text{uptake of DMC39pN into cells with low expression of integrin}]$ with the ratio, $[\text{uptake of DMCrigid into cells with high expression of integrin}]/[\text{uptake of DMCrigid into cells with low expression of integrin}]$.

2. The authors examined delivery of cargo using DMC immobilized on a solid support that grabs the DMC structure for efficient dissociation of FB by the stretching force. However, DMC is freely suspended in actual situation during drug delivery. Can FB still detached from

DMC when DMC is suspended in cell culture media?

3. The cell-specific drug delivery based on mechanical force-responsiveness is conceptually acceptable in the experimentally defined environment as shown in the manuscript.

However, I doubt that the strategy is practically feasible in vivo, because DMC should overcome in vivo barriers such as nanoparticle clearance mechanisms, nuclease degradation, and opsonization before reaching the target tissue containing the target cells.

In vivo demonstration of the feasibility of the DMC-based delivery strategy would be necessary to justify its potential in practice.

Reviewer #1 (Remarks to the Author):

General comment: The work by Velusamy et al uses previously developed DNA tetrahedral designs that are functionalized with cargo (dextran or oligonucleotide) that is delivered (uptaken by cell) in a force-sensitive manner, allowing specific delivery to cells by force-induced rupture of the tetrahedron induced by the cell surface receptors. The authors suggest promising clinical applications based on specifically targeting phenotype of cells expressing integrin adhesion receptors.

In my opinion, **the work is original and of interest to the broad community of readership**. The theoretical modeling results, obtained by oxDNA simulations, are supported by experiments, where authors demonstrate successful delivery of the cargo from the force-ruptured tetrahedrons.

I do have several questions I would like the authors to explain about the design of the nanostructures and their computational validation. Provided they can provide a satisfactory explanation, I recommend publication.

Comment 1: For testing the force-rupture in oxDNA model, is it correct that in all cases the authors anchored two different points of the structure (cRGD-attachment point and methyl-tetrazine attachment to the surface). If used in the bulk, these structures would be interacting with targeted cells without the methyltetrazine attachment to the surface. Do the authors consider that the simulation studies would still be representative of the way that the force would be applied during mechanical pulling of the surface receptor in that case?

Response: Although the present experimental and modeling work is focused on *in vitro* studies of cargo delivery, we envision *in vivo* applications where the DMCs are tethered to ECM (see figure on right). ECM-targeting is a new area of focus in drug delivery currently¹. When used in this way, the methyltetrazine group would be replaced with an ECM-binding group. This can be achieved with ECM binding ligands such as fibronectin binding peptides and ECM-specific antibodies²⁻⁴. We have elaborated further about translating our DMC designs for *in vivo* applications in supplementary note 1.

Fig 1. DMCs tethered to ECM matrix of diseased tissues. DMCs can be decorated with ECM binding motifs to localize them to specific tissues. Once anchored the DMCs can be pulled open by receptors that bind to the DMC's ligands to release the encapsulated drug.

Comment 2: The authors use k_{eff} spring constant to calculate the force that the structure experiences during pulling. Is there a reason to calculate k_{eff} , rather than obtain the force acting on the first and second anchored nucleotide respectively.

Response: The force acting on the first (cRGD) and second (methyl tetrazine) traps in series have been combined as a single *effective* trap to ensure that the force estimate is derived from the combined extension of both traps. This is commonly done in literature⁵. For the benefit of the reviewer, we have attached the force calculation from the individual traps and show that they yield the same result as having an effective trap.

Fig 2. Force estimation with one or two traps. Force-extension curves of the DMC_{29pN} calculated using either one or both traps. The data (gray shaded) has greater fluctuations when only one trap is used and hence k_{eff} was used in all cases.

Comment 3: The authors studied the rupture force by moving harmonic traps at constant speed and measuring the force induced on the system. They only have 1 trajectory per system studied, so it might not necessarily be representative and might be a very rough estimate. Additionally, it is not clear if the rupture force could be in fact lower, but since the pulling can be too fast, the actual rupture force can be over-estimated. It would be interesting if for one system (e.g., 39 pN cage), authors could try pulling and half the speed and twice the speed and see if they still obtain the same rupture force.

Response: We agree with the reviewer's assessment that a lower pulling rate will show lower force thresholds for unfolding. It must be noted that decreasing the loading rate leads to longer computational times with diminishing returns on the accuracy of the rupture estimate. For example, in our hands a loading rate of 14,000 nm/sec simulation of a typical DMC already requires 4 days of continuous calculations on our cluster. We would like to clarify to the reviewer that the statement that "only one trajectory has been studied for the DMC" is incorrect. We showed in the **Supplementary Figure 4** that the force oriented at 6 different angles leads to similar rupture forces. To further demonstrate the established dependence of rupture force on loading rates, we have attached force extension data at 3 different loading rates (see below), which is now included in the supplementary information (**Supplementary Figure 2**).

Fig 3. DMC_{29pN} force estimation under different loading rates. The FB strand on the DMC_{29pN} is pulled at different loading rates along the z-axis in oxDNA. The data for 1.4×10^6 nm/s (red), 1.4×10^5 nm/s (blue), 1.4×10^4 nm/s (yellow) loading rates were smoothed using an exponential smoothing function and plotted as force-extension graph.

Comment 4: I am not sure I understand the claim that DMC_{rigid} can withstand forces up to 500 pN. Overstretching force for DNA duplex is around 70 pN or so, so this number seems to like large overestimate. In the provided figure of stretched DMC_{rigid} , the ssDNA seems to be completely extended, so it is difficult to me how it can still remain part of the structure and not release the cargo?

Response: Indeed, the overstretching transition for a DNA duplex is at 70 pN. However, the force in the DMC_{rigid} is transmitted through a 14 bp dsDNA segment that dehybridizes to form a ssDNA segment. The rest of this DNA strand is "strain-free". Hence, denaturation does not lead to dissociation of the DMC. The following figures should help clarify this point to the reviewer.

Fig 4. DMC_{rigid} remains unperturbed under high forces. DMC_{rigid} is pulled between two points of the same strand which are 14 bp apart and rupture due to over stretching. The force is transmitted only through the 14 bp ssDNA segment (red shaded) after rupture and the rest of the structure experience no force. This allows the encapsulated cargo to remain within the DMC even under high forces.

Comment 5: For clinical application, would it be an issue that the DNA origami can get absorbed by most cells by endocytosis, and not just by the specifically targeted cells with integrin receptors on their surface?

Response: The reviewers raise a valid point that DMCs can be taken nonspecifically by cells in an *in vivo* setting. Indeed, DMCs have been used in this way for siRNA and small molecule drug delivery. For our clinical applications, we would tether the DMCs to ECM using homing ligands. Then the DMCs can be pulled open by specific receptors and release a larger quantity of therapeutics in close proximity to the cells which will reach concentrations close to therapeutic doses compared to nonspecific uptake by cells.

Comment 6: The authors use oxDNA model to study the function of the system, yet no article that describes the model is cited. Authors should provide the citations along with explanation which version and implementation of the model they used.

Response: We thank the reviewers for pointing this out and have added citations for the oxDNA2 model that was extensively used in the simulation. We have further described the version and implementation of oxDNA in the supplementary information and methods section.

Comment 7: The figure resolution is rather poor and should be increased in the final version of the manuscript.

Response: We have updated the figures and we would like to thank the reviewers for pointing this out.

Reviewer #2 (Remarks to the Author):

General comment: In the manuscript by Salaita et al. the authors describe a mechanoresponsive DNA nano-material for delivering cargo to cells via programmed degradation induced by engagement of cell surface receptors that serve to pull apart the DNA nanostructure and release its encapsulated cargo. *It is a very interesting concept – and is shown to work well by their data.* Its translational promise is not immediately apparent, but the idea is shown to be feasible through these studies and that it could be researched further to enable its potential for therapeutic application.

Response: Thank you for finding the work of interest. We now address the translational potential in a new **supplementary note 1** (see response to Reviewer #1 comment 1).

Comment 1: The anisotropy data in Fig 3B is of how many runs? Error bars or a description of the number of trials should be included.

Response: We thank the reviewer for bringing this to our attention. We have added the number of replicates as well as the error bars which depict SEM.

Comment 2: In the experiments depicted in figure 3f-j why was a 2-hour time point chosen? Do the ratios of fluorescence uptake expected to change with time?

Response: The 2-hour time point was chosen based on prior work by our lab and others that measured cell generated forces on a substrate. In these past studies it was shown that tension signal is detected 30-90 minutes after cell seeding. Given that the mechanism of DMC release of cargo is force-triggered, we used similar time points. To make this point clear to readership, we added the following statement to pg. 9 line 10:

"We performed the force-induced uptake measurement at the 2 hr time point because prior work showed DNA tension signal 30-90 min after cell seeding on a substrate^{6,7}"

Comment 3: Regarding the comment "this approach could be used to target specific cells that display high traction forces which is often associated with invasive cancer cells" with regards to the motivation for mechanoresponsive delivery of RNA to cells – would this play out in vivo or would this be solely in vitro? Could the mechanism of how this might be possible in the complex environment of many cells be explained?

Response: We appreciate the reviewer's question on scope of the work. We have now addressed this question in **supplementary note 1**.

Comment 4: In figure S9, the y axis title is confusing as it says "change" (similar to figure 4e in the main paper)– usually it is a plot of total mRNA expression. Written as change it seems confusing that the 50nM sample achieved a 23% expression but a 200nM treatment had 63% expression.... may just be the way it is written; it is somewhat hard to interpret.

Response: We thank the reviewer for pointing this out and we have changed our figure legends in the SI and the Fig 4e to be consistent.

Comment 5: For the knockdown experiments the forward and reverse primer sequences used for both the reference gene and the HIF1a target should be included in the sequences table in the supplementary information.

Response: We provided the primer sequences of both HIF1 α and 18S in **Supplementary table 2**.

Comment 6: What is the proposed mechanism of endosomal escape of the ASO tethered to the RGD ligand? Are there concerns about the stability of the oligo once released from the DMC?

Response: We hypothesize that the RGD, ASO-conjugated DMC-DNA gets uptaken through integrin mediated endocytosis and eventually dissociated from the integrin complex into the cytosol. The ASO is fully phosphorothioate modified with some LNA modifications, so there are no stability concerns for the oligonucleotide cargo after it is released from the DMC.

Comment 7: Figure quality overall is low (for example, figure 1 and others are very blurry/low resolution).

Response: The lower quality is likely due to compression when the pdf was generated by the Nature system. We will upload full resolution figures in this resubmission.

Comment 8: Line 38 in the introduction's first paragraph appears to be missing a word – it looks like something is missing grammatically.

Response: We have re-written this sentence to make it clearer:

"The motivation for using a specific magnitude mechanical force as a cue comes from quantitative measurements of forces generated by many classes of receptors such as Integrins^{8,9}, T cell^{10,11}, and B cell receptors^{12,13}, Notch^{14,15} among others^{6,16}."

Comment 9: Figure 3a, the schematic showing the encapsulated cargo (green trace) should be better aligned with the green line – looks like its pointing to the blue line.

Response: We have rectified this in our current draft of the manuscript, and we appreciate the reviewer's attention to detail.

Reviewer #3 (Remarks to the Author):

General comment. The manuscript by Arventh Velusamy et al. reports a novel method to release drug molecules encapsulated in DNA mechanocapsules (DMC) by using mechanical forces generated by cells. Considering the range of forces generated by different receptors, the authors designed and simulated different tetrahedral DMCs displaying controlled rupture forces. An extensive proof of concept study is presented to validate the functionality of DMCs under forces expressed by integrin receptors, the encapsulation, release and uptake of drug cargos. Finally, force-responsive RNA knockdown is demonstrated in cells by using functionalized DMCs.

The main novelty and contribution of this work is the demonstration of the possibility to encapsulate and release a wide variety of cargos by mechanically forces expressed by cell receptors. This type of cell targeting opens a multitude of research lines in cell biology, biochemistry and biophysics potential application to biomedicine, immunology, and pharmacology.

In general, the paper is well written, and the concepts clearly presented. However, I have two main concerns regarding the DMC characterization: only a loading rate was considered in simulations and no DMC characterization by force spectroscopy. Therefore, I recommend publication pending on Major Revision. Details supporting these concerns and further comments follow.

Comment 1: The DMCs design is clever, and the authors elegantly show the possibility to tune the rupture force by changing the RDG anchoring points and the force bearing strand lengths. The response to force orientation is also analyzed, concluding that it does not influence unfolding and DMC_{39pN} release encapsulated cargo for forces bigger than 39 pN. This value is related to the forces exerted by integrin receptor on the RDG ligand, thus making it possible to trigger the release of an encapsulated cargo by a cell in contact with DMC. While this rationale is helpful, my concern is how the value 1.4×10^4 nm/s of the pulling rate was chosen. With the stiffness $k = 5.71$ pN/nm, given in Methods 7.1, a loading rate of 8×10^4 pN/s is obtained.

Response: The traps are moved at a rate of 5×10^{-8} (length per unit of time in oxDNA units). This rate can be converted into SI units as shown here:

$$\text{Loading rate} = \frac{5 \times 10^{-8} \times 0.8518 \text{ nm}}{3.03 \times 10^{-12} \text{ s}} = 1.4 \times 10^4 \text{ nm/s}$$

We have added this calculation to our methods section to add more clarity.

Comment 2: Moreover, since it is known that the loading rate represents a critical variable in stiffness sensing, I would expect a force analysis as a function of the loading rate with examples taken in a wide range (e.g. 10^2 – 10^5 pN/nm). The authors mention in the text (Conclusions) that “*the molecular loading rates are currently unknown*”. Probably they refer to the specific RDG – integrin interaction. However, there is literature in which the loading rate values for integrin interacting with ligand are at least predicted and discussed if not measured (e.g. fibronectin). Few examples: Andreu 2021, Nat Comm: <https://www.nature.com/articles/s41467-021-24383-3>, Amar 2020, Biochem Biophys Res Comm: <https://doi.org/10.1016/j.bbrc.2020.01.149>, Kolasangiani 2022, Cells <https://doi.org/10.3390/cells11223584>, Li 2003, Biophys J: [https://doi.org/10.1016/S0006-3495\(03\)74940-6](https://doi.org/10.1016/S0006-3495(03)74940-6). Considering the importance of the loading rate also for the rupture force, I recommend to run new simulations and discuss the results obtained.

Response: As a response to reviewer 1 and 3 we have run new simulations with loading rates and the results indicate that faster loading rates lead to higher force threshold in line with published literature (see above, Fig 3).

Comment 3: An explanation on the elastic constants (both 11.42 pN/nm) is missing in methods and hence it is requested.

Response: These values are based on literature precedent as was done in Engel *et al.*, 2018; 10.1021/acsnano.8b01844. Both the traps were set to be at the same stiffness to simplify the formula for two springs in series. For two springs with same elastic constants in series k_{eff} becomes $k/2$.

Comment 4: Fig. 1b: does the force jump have a meaning?

Response: The force drop in Fig 1b represents the rupture of the DMC leading to the relaxation of the traps that were extended until that point in the simulation. This relaxation leads to the force dropping to zero. This is now better explained in the main text – Page 3 Line 15:

“The plot in Fig. 1B shows the trajectory of DMC_{39pN} that undergoes significant deformation prior to the release of FB strand at 39 pN followed by a force drop due to rupture.”

Comment 5: Fig. 1e: as regards the plot of the pore radii in fig. 1e, the figure is not clear (not visible e.g. behavior for 44 pN). Is the range defined by min and max? Maybe representing only, the mean + std values at several instants would help to show better the pore size variation under force.

Response: Because cargo leakage is an irreversible process, the maximum pore size at a given time point is the most relevant parameter to predict cargo release. We have updated the figure to plot the maximum pore size (rather than the max-min range).

Comment 6: Comments on “Synthesis and characterization of DMCs” section - The synthesis and deposition of DMCs on substrate are clearly presented. However, the images (fig 3 b,c,d) showing quenched and unquenched DMCs are too small and poorly explained to be helpful. For instance, the unquenched fluorescence intensity seems to vary a lot. Does this mean that the deposition is not uniform? How is the intensity measured for QE? Averaged over the area of the image? Please explain and comment.

Response: We have resized the figures to make these clearer. We also updated the main text and captions to better explain the reason the images are included and interpretation of the data. The variance of the images is simply due to the scaling of the LUTs and also because each experiment used different concentrations of soluble DMCs. Quenching efficiency is given by the formula:

$$Q.E. = 1 - \frac{I_{\text{quenched}}}{I_{\text{unquenched}}} \times 100$$

The intensity was reported based on the average values from three independent measurements of three preparations. Each surface intensity was analyzed from the whole image from multiple regions. We would like to keep these images in the main as these are useful to orient the reader and offer the raw data from which the analysis is provided.

Comment 7: I would also suggest that these images go to SI, with a better explanation and proper size. As regards the high DMCs density on the substrate, I wonder how this condition can be handled in vivo i.e. for cells and ECM. Please comment (more than is already done in conclusions) in terms of applicability and density required.

Response: Please see **supplementary note 1**.

Comment 8: Fig. 2c Time (m)? maybe (min). Same for fig 3b

Response: We have changed these legends in the figures to Time (min) as per the reviewers' suggestions.

Comment 9: The examples shown for NIH3T3 cells and three types of breast cancer cell are illustrating well the

potential of DMC activation by cellular force, although the associated figures 2e-i can be improved:- the images (tension and paxilin) are too small and the correlation, if any, with RICM not explained.

Response: We have resized the figures to increase clarity. We apologize for neglecting to define RICM and relating it to the fluorescence images. This is now included in the figure captions.

Comment 10: The min and max scale of color bars are not indicated (i.e., 5000 counts of max?) and hence not too relevant.

Response: We include the min and max scale of each fluorescence image for transparency and so that the reader can compare across images that are scaled differently. We are not sure if the reviewer would like us to keep this information or to remove it.

Comment 11: Is there an explanation for the tension shown in fig 2g for MDA-MB-231 (internal ring)? Is there a traction additional to the traction at the cell edge?

Response: The ring pattern has been documented in past works using DNA tension probes^{7,16}. Cells usually display a ring of tension signal at the lamellipodial edge due to actin bundles accumulating at the cell edge with a puncta at the cell center. The ring of tension is highly dynamic and it follows the edge of the cell as it spreads. Since the DMCs report a history of cellular tension, the two rings observed could likely be due to the spreading process of the cell and the magnitude of applied traction forces during different spreading phases.

Comment 12: The graph in fig 2f is showing two-line profiles from 2e, but the line profiles are clearly not correspondent. Moreover, colocalization is not the appropriate term for this type of images (diffraction limited) where the structure sizes are much bigger than the molecular size. Thus fig. 2f does not help illustrating colocalization and should be removed or replaced with profiles taken at a high enough resolution.

Response: The reviewer makes a good point and this is a common pitfall in literature. But in our case, we are simply validating the claim that the tension signal is associated with markers of focal adhesions. We are not claiming a specific protein-protein interaction. Rather, we are simply confirming that tension is associated with focal adhesions which by the way are micron scaled when matured.

Comment 13: SI Fig 7 is cut at left, in c there is no error bar for 50 nM.

Response: We have corrected the figures so that they are not cropped at the left margin. The error bars are too small to be shown in the graphs. We have added the error values in the accompanying text.

Comment 14: Fig. 2g and h: why qDMC_{27pN} and not 39pN? What does it mean they showed greatest response for traction forces? The spread is related to substrate stiffness. Please comment more.

Response: We tested multiple DMCs with different response thresholds and intentionally showed the DMC that produced the greatest difference. To address this point, we show here the cell response with DMC_{39pN}.

Comment 15: MCF-7 image is very unclear, due to a bright spot- is this spot from measurement or is it an artefact?

Response: The spot is an artefact in the surface (left image panel), and we have replaced the image with a better representative image (right image panel).

Comment 16: Main concern: Although the experiments presented in this section confirm the DMC activation by cellular forces, a methodologic step is missing after the simulation of the activation forces and before the experiments with cells: the characterization of DMC activation by force spectroscopy. In my opinion this is an important step which is required to complete DMC characterization.

Response: Force spectroscopy validation of the modeling would indeed be desirable, however the forces required to rupture DMC are beyond the range of most optical tweezer and magnetic tweezer systems. Hence the force spectroscopy would require AFM – and this is currently not available to us. Moreover, oxDNA model has already been validated and shown to agree with several experimental data including force spectroscopy^{17–22}.

Comment 17: Comments on Cargo encapsulation and in vitro evaluation of DMCs and force-responsive RNA knockdown. This is a nice proof of concept of drug encapsulation and delivery by using DMC and cellular forces. Just as a curiosity, I wonder if a 44% encapsulation efficiency can be considered sufficient and in which conditions it can be improved. Please comment.

Response: The reported 44% encapsulation is specifically for kinetically trapping fluorescent-Dextran molecules when annealing the DMC. The encapsulation efficiency of the cargo can change depending on the nature of the cargo such as size, charge and the purification methods used. Other approaches for encapsulation, such as covalent trapping, has shown greater encapsulation efficiencies²³. Note that our ASO encapsulation efficiency is approximately 100%.

Comment 18: It is mentioned as a result that force mediated uptake is 2-fold greater than that of the controls. Is this enough and promising? Can it be enhanced? Please comment.

Response: We are confident that the force mediated uptake can be enhanced beyond the 2-fold we have described in this manuscript. We decided to leave optimization and condition screening for future works that will follow. This was a proof-of-concept for force induced drug delivery of macromolecules and we did not optimize it to get maximal encapsulation or uptake response.

Comment 19: The images in fig. 3e are too small. Reducing a bit f, g, h some space could be used to increase the size of images which otherwise are not very useful.

Response. We have taken the reviewers comments and redesigned the figure 3 to make the data look clearer.

Comment 20: The force responsive RNA knockdown: the results presented in fig. 4e are not clearly commented in the text and conclusions, please review.

Response. We have expanded our discussion of the force responsive RNA knockdown shown in figure 4e.

"The DMC_{39pN} [HIF1 α] and DMC_{39pN} [(HIF1 α)₂] liberate one and two ASOs respectively. The release occurs in a force-induced manner which will likely be endocytosed by integrin recycling pathways^{24,25} as the ASOs are covalently linked with the integrin-binding RGD peptide (Fig 4B). To control for non-specific release, DMCs were conjugated with ASOs linked to a non-FB strand (DMC_{inert} [HIF1 α]) which would not be released under force (Fig 4C). To maintain identical surface densities of DNA and RGD the DMCs were formulated as a binary mixture: 1) DMC_{39pN} [(HIF1 α)₂] + DMC_{inert} 2) DMC_{39pN} [HIF1 α] + DMC_{inert} 3) DMC_{inert} [HIF1 α] + DMC_{39pN}. HeLa cells were cultured on these three DMC grafted surfaces with an initial 6-hour serum starvation phase followed by 10% serum addition and growth. The mRNA levels of these cells were then quantified after 24 hours using RT-qPCR. The force responsive DMC_{39pN} [(HIF1 α)₂] and DMC_{39pN} [HIF1 α] produced a 37 \pm 8% and 20 \pm 7% knockdown of HIF1 α RNA levels, respectively, whereas the unresponsive DMC_{inert} [HIF1 α] had almost no change in mRNA levels (~5% reduction) compared to cells cultured on a DMC_{39pN} (Fig 4D)."

Comment 21: The argumentation on the reduction of the total concentration of drug is not clear, and hence not convincing. I suggest to bring the info from SI to main text and make the discussion clearer.

Response: We have placed this in the SI in the interest of the word limit constraints in the article.

Reviewer #4 (Remarks to the Author):

General comment: The authors describe DNA mechanocapsules (DMC) as a mechanical force-responsive drug delivery system based on DNA tetrahedrons. The DNA tetrahedrons conjugated with RGD peptides that specifically interact integrin on cancer cell surface. The position of RGD was optimized so that the RGD-labeled strand can be stretched out and eventually dehybridized out of the DNA nanostructures by the force derived from the integrin-RGD interaction. This concept could be used to specifically release model drugs encapsulated or loaded in DNA tetrahedrons depending on the integrin expression level of cells. Although *the concept is novel and interesting*, there are issues that should be addressed before considering the manuscript as follows.

Comment 1: Force induced uptake may not only enhance the uptake of DMC_{39pN} into the cells with high integrin expression but also facilitates uptake of DMC_{39pN} into the cells with low integrin expression. To address this concern, the authors need to examine whether [uptake into target cell]/[uptake into non-target cell] ratio is indeed improved. Also, they have to compare the ratio, [uptake of DMC_{39pN} into cells with high expression of integrin]/[uptake of DMC_{39pN} into cells with low expression of integrin] with the ratio, [uptake of DMC_{rigid} into cells with high expression of integrin]/[uptake of DMC_{rigid} into cells with low expression of integrin].

Response: The reviewer makes a valid point that cells with low integrin or force levels can also have collateral uptake from nearby cells that have high levels of force/integrin expression. To demonstrate that collateral uptake occurs minimally, we have performed experiments with a co-culture of high force and low force phenotype cells. We have added these data to our manuscript in figure 3i, 3l as well as supplementary figure 9.

Briefly, we co-cultured MEF cells expressing GFP-tagged vinculin and MEF cells with vinculin knocked out. When these cells were added to a surface functionalized with DMC_{39pN} [Dex_{647N}] we found out that MEF cells that express vinculin (high force phenotype) had 2-fold higher uptake compared to MEF cells without vinculin (low force phenotype). When the two MEF variants were added to separate surfaces a 3-fold difference in uptake was observed. The slight reduction in differential uptake between the mixed group could be due to population gating based on GFP fluorescence. A small population of MEF cells with vinculin were observed to have lower levels of GFP expression. These cells show up on the flow cytometer along with the vinculin knockout population which could result in them having higher fluorescence values. This in turn could reduce the observed differential uptake in the mixed population than when the high force and low force population were separately cultured on different surfaces.

Comment 2: The authors examined delivery of cargo using DMC immobilized on a solid support that grabs the DMC structure for efficient dissociation of FB by the stretching force. However, DMC is freely suspended in actual situation during drug delivery. Can FB still be detached from DMC when DMC is suspended in cell culture media?

Response: We understand the reviewers' concerns and we propose that the DMCs be anchored using ligands that bridge between two force producing molecules such that the force threshold unravels the origami to release the contents of the DMC. This is now addressed in **supplementary note 1**.

Comment 3: The cell-specific drug delivery based on mechanical force-responsiveness is conceptually acceptable in the experimentally defined environment as shown in the manuscript. However, I doubt that the strategy is practically feasible *in vivo*, because DMC should overcome *in vivo* barriers such as nanoparticle clearance mechanisms, nuclease degradation, and opsonization before reaching the target tissue containing the target cells. *In vivo* demonstration of the feasibility of the DMC-based delivery strategy would be necessary to justify its potential in practice.

Response: We understand the concerns raised by the reviewers and we agree that all these barriers plague most drug delivery strategies. Regardless DMCs are constructed out of DNA tetrahedrons that have significant nuclease compared to similar DNA based systems which can further be improved by adding modifications to DNA such as PS, LNA. Further, DNA tetrahedrons are reported to be cleared slowly from compared to duplex systems. Hence, we believe that DMC is suitable for *in vivo* drug delivery applications.

References

1. Huang, J. *et al.* Extracellular matrix and its therapeutic potential for cancer treatment. *Signal Transduct. Target. Ther.* **6**, 1–24 (2021).
2. Arnoldini, S. *et al.* Novel peptide probes to assess the tensional state of fibronectin fibers in cancer. *Nat. Commun.* **8**, 1793 (2017).
3. Saw, P. E. *et al.* Aptide-conjugated liposome targeting tumor-associated fibronectin for glioma therapy. *J. Mater. Chem. B* **1**, 4723–4726 (2013).
4. Lo, K.-M. *et al.* huBC1-IL12, an immunocytokine which targets EDB-containing oncofetal fibronectin in tumors and tumor vasculature, shows potent anti-tumor activity in human tumor models. *Cancer Immunol. Immunother.* **56**, 447–457 (2007).
5. Engel, M. C. *et al.* Force-Induced Unravelling of DNA Origami. *ACS Nano* **12**, 6734–6747 (2018).
6. Zhang, Y., Ge, C., Zhu, C. & Salaita, K. DNA-based digital tension probes reveal integrin forces during early cell adhesion. *Nat. Commun.* **5**, 5167 (2014).
7. Ma, R. *et al.* DNA probes that store mechanical information reveal transient piconewton forces applied by T cells. *Proc. Natl. Acad. Sci.* **116**, 16949–16954 (2019).
8. Liu, Y. *et al.* Nanoparticle Tension Probes Patterned at the Nanoscale: Impact of Integrin Clustering on Force Transmission. *Nano Lett.* **14**, 5539–5546 (2014).
9. Zhang, Y. *et al.* Platelet integrins exhibit anisotropic mechanosensing and harness piconewton forces to mediate platelet aggregation. *Proc. Natl. Acad. Sci.* **115**, 325–330 (2018).
10. Liu, Y. *et al.* DNA-based nanoparticle tension sensors reveal that T-cell receptors transmit defined pN forces to their antigens for enhanced fidelity. *Proc. Natl. Acad. Sci.* **113**, 5610–5615 (2016).
11. Ma, V. P.-Y. *et al.* The magnitude of LFA-1/ICAM-1 forces fine-tune TCR-triggered T cell activation. *Sci. Adv.* **8**, eabg4485 (2022).
12. Wang, J. *et al.* Profiling the origin, dynamics, and function of traction force in B cell activation. *Sci. Signal.* **11**, eaai9192 (2018).
13. Ketchum, C. M. *et al.* Subcellular topography modulates actin dynamics and signaling in B-cells. *Mol. Biol. Cell* **29**, 1732–1742 (2018).
14. Narui, Y. & Salaita, K. Membrane Tethered Delta Activates Notch and Reveals a Role for Spatio-Mechanical Regulation of the Signaling Pathway. *Biophys. J.* **105**, 2655–2665 (2013).
15. Wang, X. & Ha, T. Defining Single Molecular Forces Required to Activate Integrin and Notch Signaling. *Science* **340**, 991–994 (2013).
16. Brockman, J. M. *et al.* Live-cell super-resolved PAINT imaging of piconewton cellular traction forces. *Nat. Methods* **17**, 1018–1024 (2020).
17. Ouldrige, T. E., Šulc, P., Romano, F., Doye, J. P. K. & Louis, A. A. DNA hybridization kinetics: zippering, internal displacement and sequence dependence. *Nucleic Acids Res.* **41**, 8886–8895 (2013).
18. Srinivas, N. *et al.* On the biophysics and kinetics of toehold-mediated DNA strand displacement. *Nucleic Acids Res.* **41**, 10641–10658 (2013).
19. Romano, F., Chakraborty, D., Doye, J. P. K., Ouldrige, T. E. & Louis, A. A. Coarse-grained simulations of DNA overstretching. *J. Chem. Phys.* **138**, 085101 (2013).
20. Kriegel, F. *et al.* The temperature dependence of the helical twist of DNA. *Nucleic Acids Res.* **46**, 7998–8009 (2018).
21. Snodin, B. E. K., Schreck, J. S., Romano, F., Louis, A. A. & Doye, J. P. K. Coarse-grained modelling of the structural properties of DNA origami. *Nucleic Acids Res.* **47**, 1585–1597 (2019).
22. Khara, D. C. *et al.* DNA bipedal motor walking dynamics: an experimental and theoretical study of the dependency on step size. *Nucleic Acids Res.* **46**, 1553–1561 (2018).
23. Erben, C. M., Goodman, R. P. & Turberfield, A. J. Single-Molecule Protein Encapsulation in a Rigid DNA Cage. *Angew. Chem. Int. Ed.* **45**, 7414–7417 (2006).
24. Bridgewater, R. E., Norman, J. C. & Caswell, P. T. Integrin trafficking at a glance. *J. Cell Sci.* **125**, 3695–3701 (2012).
25. Moreno-Layseca, P., Icha, J., Hamidi, H. & Ivaska, J. Integrin trafficking in cells and tissues. *Nat. Cell Biol.* **21**, 122–132 (2019).

REVIEWER COMMENTS

Reviewer #1 (Remarks to the Author):

The authors have satisfactorily addressed my comments and I recommend publication after the following minor revision: Regarding the point raised about the rupture force estimation, I suggest it should be explicitly mentioned in the text that the rupture force estimation is based on a particular loading rate used in the simulation, and that lower loading rate would lead to somewhat lower rupture force, along with reference to the supplementary figure S2G provided.

Reviewer #2 (Remarks to the Author):

The reviewers have sufficiently answered my concerns and updated the manuscript in response to the issues described.

It is suitable for publishing.

Reviewer #3 (Remarks to the Author):

Thank the authors for their answers and revision.

The manuscript has been improved and I recommend it for publication.

Minor comments:

1. My question was: “how the value 1.4×10^4 nm/s of the pulling rate was chosen?”, not how it was calculated. I.e. explain the reason for choosing the value 1.4×10^4 nm/s and not another value.

2. Simulations have been run with additional (two) pulling rates, indicating higher rupture forces for higher pulling rates, as expected from theory and experiments in literature. While I appreciate the effort for simulating different pulling rates, I do not understand why authors have not considered also lower pulling rates.

Corroborated with the stiffness value chosen in the simulation model, the two new pulling rates: 1.4×10^5 and 1.4×10^6 nm/s give values of the loading rates higher than the range I could find in literature (see references I already indicated). In fact, even from the point of view of the pulling rate alone, a value of 1.4 mm/s can hardly be imagined as realistic for these molecular processes.

Moreover, having an estimate of the rupture force versus loading rate could provide also useful information on the dissociation constant at zero force; it would be interesting if authors can/want comment on this point in the paper.

3. Authors answered to comment 16:

“Force spectroscopy validation of the modeling would indeed be desirable, however the forces required to rupture DMC are beyond the range of most optical tweezer and magnetic tweezer systems. Hence the force spectroscopy would require AFM – and this is currently not available to us. Moreover, oxDNA model has already been validated and shown to agree with several experimental data including force spectroscopy^{17–22}”.

I agree that a full range validation (from low to high loading rates) would require both optical tweezers and AFM techniques and many experiments and time. In my opinion, the validation of oxDNA by force spectroscopy for other experimental situations partially justifies the lack of experimental confirmation here. I recommend authors write a sentence to justify their choice in the paper.

4. Answer to comment 10: “We are not sure if the reviewer would like us to keep this information or to remove it”.

Keep it.

Reviewer #4 (Remarks to the Author):

I do not think the manuscript suitable for publication in this journal yet as the authors have not addressed the concerns.

Regarding Comment 1, the authors need to examine the collateral uptake of DMCRigid [Dex647N] as well as that of DMC39pN [Dex647N] in GFP-Vin and Vin-KO cells. Whether the selectivity for GFP-Vin over Vin-KO (ca. 1.7-fold) is superior to that of DMCRigid [Dex647N] should be examined to properly evaluate the target cell specificity by their force-responsive drug delivery system.

Regarding Comment 3, I agree with the authors that all the drug delivery systems face the in vivo barriers. However, most of the drug delivery strategies reported have demonstrated their potential by showing feasibility in an in vivo model or at least an in vivo-mimicking environment. I do not see any evidence in the revised manuscript supporting such feasibility. By the way, was the cellular uptake experiment performed in the presence of 10% FBS? No information on the media condition is indicated in the manuscript.

Reviewer #1 (Remarks to the Author):

The authors have satisfactorily addressed my comments and I recommend publication after the following minor revision.

Regarding the point raised about the rupture force estimation, I suggest it should be explicitly mentioned in the text that the rupture force estimation is based on a particular loading rate used in the simulation, and that lower loading rate would lead to somewhat lower rupture force, along with reference to the supplementary figure S2G provided.

Response: We appreciate the reviewer's suggestion and the updated manuscript states that the rupture force is loading rate dependent. Note that the current loading rate required approximately one week of continuous computation time on the Emory computational core (Cherry Emerson Center for Scientific Computation). As the computational time scales linearly with loading rate, running simulations at slower loading rates is currently impractical (see Methods). Regardless, our estimated rupture force values are likely accurate as values obtained for dsDNA rupture and shearing match with that of experiments.

New text: "Note that rupture force is highly loading rate dependent as has been well documented in the literature^{1,2}, and lower loading rates will lead to dampened rupture force values. The loading rate used for DMC simulations was 1.4×10^4 nm/s, which was found to be appropriate as we benchmarked the rupture of dsDNA using oxDNA against that of force spectroscopy values and we see general agreement, thus validating the simulations (Supplementary Fig. 2G, H). Using this loading rate, modelling showed that the three DMCs displayed rupture forces of 27.0 ± 0.6 pN, 39.0 ± 0.5 pN and 43.5 ± 0.5 pN (Fig. 1C)."

Reviewer #2 (Remarks to the Author):

The reviewers have sufficiently answered my concerns and updated the manuscript in response to the issues described. It is suitable for publishing.

Reviewer #3 (Remarks to the Author):

Thank the authors for their answers and revision. The manuscript has been improved and I recommend it for publication.

Minor comments:

Comment 1: My question was: "how the value 1.4×10^4 nm/s of the pulling rate was chosen?", not how it was calculated. i.e., explain the reason for choosing the value 1.4×10^4 nm/s and not another value.

Response: We would like to thank the reviewer for clarifying their question. Lower loading rates are certainly more desirable as the biological loading rates are likely at rates that are significantly lower. However, we have finite computational and financial resources at our disposal and the trade-off between computational cost/time and quality of the prediction led us to this loading rate used in this present paper. For example, at the 1.4×10^4 nm/s loading rate, a single simulation on the Emerson Computational Core at Emory required approximately one week of continuous calculation. The manuscript includes 30 – 40 simulations and hence using lower loading rates is impractical. Moreover, we do know that at this loading rate oxDNA predicts a DNA duplex shearing force of ~ 60 pN^{3,4} which is $\sim 10\%$ greater than the experimentally observed rupture force for short (21mer) duplexes. This suggests that the 1.4×10^4 nm/s loading rate still offers meaningful rupture forces that are fairly accurate assuming that the experiments provide the ground truth. We have updated the text to clarify this point. Also, please see response to reviewer #1 which addresses the same point.

Comment 2: Simulations have been run with additional (two) pulling rates, indicating higher rupture forces for higher pulling rates, as expected from theory and experiments in literature. While I appreciate the effort for simulating different pulling rates, I do not understand why authors have not considered also lower pulling rates.

Corroborated with the stiffness value chosen in the simulation model, the two new pulling rates: 1.4×10^5 and 1.4×10^6 nm/s give values of the loading rates higher than the range I could find in literature (see references I already indicated). In fact, even from the point of view of the pulling rate alone, a value of 1.4 mm/s can hardly be imagined as realistic for these molecular processes.

Moreover, having an estimate of the rupture force versus loading rate could provide also useful information on the dissociation constant at zero force; it would be interesting if authors can/want comment on this point in the paper.

Response: Please see our response to Reviewer 1, comment 1, and the response to your first comment. Additionally, our loading rate values are in line with those used in prior publications. See for example:

Paper 1. "Force-Induced Unravelling of DNA Origami", *ACS Nano* **12**, 6734–6747 (2018). Here the authors used a number of different loading rates, but the slowest used was 5.6×10^7 nm/s.⁵

Paper 2. "Coarse-grained simulations of DNA overstretching" Coarse-grained simulations of DNA overstretching. *J. Chem. Phys.* **138**, 085101 (2013). Here the manuscript models shearing of simple DNA duplexes which are considerably less computationally intensive than DMC. Nonetheless, the team uses a loading rate that is as low as 2.8×10^4 pN/s for dsDNA which translates into 2.8×10^2 nm/s.⁶

It must be noted that the simulation times also scale up with larger DNA structures in the oxDNA model. Finally, we have benchmarked this loading rate in oxDNA by estimating the shearing forces for a 21 bp DNA duplex (see below). Our results are close to the experimentally determined dsDNA shearing force from magnetic tweezers.⁷

Fig 1: oxDNA benchmarks a) DNA duplex rupture force estimated using oxDNA under the same loading rate of 1.41×10^4 nm/s. b) Data from Mosayebi et al. (4) comparing the fit of oxDNA model (squares) to the experimental data of Hatch et al. (7)

Comment 3: Authors answered to **Comment 16:**

“Force spectroscopy validation of the modeling would indeed be desirable, however the forces required to rupture DMC are beyond the range of most optical tweezer and magnetic tweezer systems. Hence the force spectroscopy would require AFM – and this is currently not available to us. Moreover, oxDNA model has already been validated and shown to agree with several experimental data including force spectroscopy^{17–22}”.

I agree that a full range validation (from low to high loading rates) would require both optical tweezers and AFM techniques and many experiments and time. In my opinion, the validation of oxDNA by force spectroscopy for other experimental situations partially justifies the lack of experimental confirmation here. I recommend authors write a sentence to justify their choice in the paper.

Response: We appreciate this comment and have amended the paper to explain why we have not performed single molecule force spectroscopy on the DMCs.

New text: “Initially, three chemically identical DMCs but differing in RGD attachment points and force-bearing (FB) strand lengths were modelled to simulate DMC responses under forces using oxDNA2. Ideally, we would employ single molecule force spectroscopy techniques to validate DMC force response, but these measurements are challenging for large libraries of structures; moreover, oxDNA rupture force predictions have been validated and shown to be accurate when compared to experimental measurements⁴. ”

Comment 4: Answer to **Comment 10:** “We are not sure if the reviewer would like us to keep this information or to remove it”.

Keep it.

Response. The calibration bars now remain in the figure as suggested by the reviewer.

Reviewer #4 (Remarks to the Author):

I do not think the manuscript suitable for publication in this journal yet as the authors have not addressed the concerns.

Regarding **Comment 1**: the authors need to examine the collateral uptake of DMC_{rigid}[Dex_{647N}] as well as that of DMC_{39pN}[Dex_{647N}] in GFP-Vin and Vin-KO cells. Whether the selectivity for GFP-Vin over Vin-KO (ca. 1.7-fold) is superior to that of DMC_{rigid}[Dex_{647N}] should be examined to properly evaluate the target cell specificity by their force-responsive drug delivery system.

Response: The DMC_{rigid} was designed as a control for the force responsive uptake experiments and it has already been evaluated for specificity in comparison to DMC_{39pN} and DMC_{inert}. Specifically, DMC_{39pN} was found to have 3-fold higher uptake compared to DMC_{rigid} in our experiments (Fig 3j). We did not use DMC_{rigid} in our further experiments since we have showed that DMC_{39pN} response is force specific and **DMC_{rigid} was not intended for cell subtype selectivity or drug delivery**. Furthermore, DMC_{rigid} probes will dissociate from the receptor or substrate under high forces. This could result in the DMC_{rigid} not releasing cargo in an *in vivo* context, rendering it ineffective for force-induced drug release. As we have shown in the manuscript, DMCs can also be engineered to release drugs at a specific piconewton force threshold which could be used to improve the selectivity between cellular subtypes. Hence, we believe we have shown sufficient evidence that DMC_{39pN} releases cargo selectively based on forces and it can in fact distinguish cells based on its mechanical phenotype.

Regarding **Comment 3**: I agree with the authors that all the drug delivery systems face *in vivo* barriers. However, most of the drug delivery strategies reported have demonstrated their potential by showing feasibility in an *in vivo* model or at least an *in vivo*-mimicking environment. I do not see any evidence in the revised manuscript supporting such feasibility.

Response: While an *in vivo* demonstration of our DMC system in the first publication is desirable, it takes substantial effort and time to translate systems to *in vivo*. The current manuscript is intended to be a **first proof of concept** communication of our system *in vitro* demonstrating **modular force-specific cargo release**.

However, to highlight DMCs *in vivo* potential we have gathered preliminary data that shows that indeed DMCs have *in vivo* potential by using homing ligands as we have discussed in **supplementary note 1**. We modified DMCs with multiple ZY-1 aptamers that bind to fibronectin through multivalent anchoring and (Figure 2b) demonstrate specific binding of the aptamers to fibronectin-coated surfaces. DNA were annealed at 5 μ M concentration and added to wells in binding buffer at 2.5 μ M for 30 mins. This was followed by three washes in wash buffer as per published protocols.⁸ DMC_{27pN} with three ZY-1 aptamers for binding had the highest intensity in fluorescence microscopy indicating specific binding. Then, DMC_{39pN} and DMC_{inert} tagged with Cy3B were encapsulated with Atto647N functionalized dextrans and anchored to surfaces. HeLa cells were seeded to these surfaces and their uptake was evaluated using flow cytometry. Enhanced uptake of dextrans was observed on DMC_{39pN} [Dex_{647N}] surfaces while DMC_{inert} [Dex_{647N}] had a 2-fold lower uptake indicating force-mediated cargo release (Figure 2e, f). The Cy3B intensity remained the same in the samples indicating minimal force-mediated dissociation and subsequent uptake of the DMC structures into the cells. Hence this preliminary data shows that DMC systems can be used for force-induced drug release *in vivo* with a wide variety of cargoes such as small molecules (conjugated to dextran), proteins and oligonucleotides. We believe further optimization of this work is best reserved for a comprehensive future therapeutic study as this would likely require multiple years of work and additional resources out of our grasp at the moment.

It should also be mentioned that our DMC systems already have a lot of potential application such as enhanced stability during prolonged force measurements *in vitro* (due to inherent stability of the DMCs), high throughput cell-tagging based on forces (fluorescent dextran uptake from DMC), force-specific manipulation of cells (ASO

Fig 2: DMC cargo delivery in an *in vivo* mimicking environment. a) Sequences of aptamer ZY-1 with thiol modification and anchor strand with three amine modifications which were converted to maleimide groups using hetero-bifunctional SMCC crosslinkers. The anchor strand was reacted with the aptamer strand in excess to obtain the ZY-1 multivalent anchor strand through thiol-maleimide conjugation. b) Demonstration of aptamer ZY-1 binding to fibronectin coated glass wells. Fibronectin-coated surfaces were added with annealed DNA at 2.5 μ M in 1x binding buffer (10% FBS, 1% BSA, 4.5g Glucose, 5 mM Mg in DPBS). The number represent average intensities. Image scale – 82 x 82 μ m². Schematic of DMC anchored on fibronectin surfaces using three ZY-1 aptamers, delivering Dextran-functionalized with Atto647N from c) DMC_{39pN} [Dex_{647N}] d) DMC_{inert} [Dex_{647N}]. DMCs were also tagged with Cy3B to monitor force-induced dissociation and uptake of the DMCs from the surface. e) 2D flow cytometry data of Dextran-Atto647N uptake (x-axis, first column in table) vs DMC-Cy3B uptake (y-axis, second column in table) under HeLa cells in DMEM (10% FBS) after 2 hours (n=3). f) Background subtracted median fluorescence intensities of Dex_{647N} uptake from DMCs by HeLa cells in (e). Each data point represents a biological replicate. **P* - 0.0159.

delivery using DMCs). We believe rapid publication of our data will inspire new avenues of research utilizing DMC systems to leverage biophysical cues to discriminate and modulate populations of cells.

By the way, was the cellular uptake experiment performed in the presence of 10% FBS? No information on the media condition is indicated in the manuscript.

Response: We have explained the methods in detail in the methods section 14.2: “The mixture of DMCs were allowed to click to TCO-PEG4 surfaces in 1x TM for about 1 hr and then washed with 10% FBS DMEM (with 1% Penicillin-Streptomycin and no phenol red). To the surfaces, MEF cells (2.5×10^4) were seeded to DMC grafted coverslips and allowed to attach for 2 hrs.”

References

1. Liu, J. *et al.* Tension Gauge Tethers as Tension Threshold and Duration Sensors. *ACS Sens.* **8**, 704–711 (2023).
2. Andreu, I. *et al.* The force loading rate drives cell mechanosensing through both reinforcement and cytoskeletal softening. *Nat. Commun.* **12**, 4229 (2021).
3. Ma, V. P.-Y. *et al.* The magnitude of LFA-1/ICAM-1 forces fine-tune TCR-triggered T cell activation. *Sci. Adv.* **8**, eabg4485 (2022).
4. Mosayebi, M., Louis, A. A., Doye, J. P. K. & Ouldrige, T. E. Force-Induced Rupture of a DNA Duplex: From Fundamentals to Force Sensors. *ACS Nano* **9**, 11993–12003 (2015).
5. Engel, M. C. *et al.* Force-Induced Unravelling of DNA Origami. *ACS Nano* **12**, 6734–6747 (2018).
6. Romano, F., Chakraborty, D., Doye, J. P. K., Ouldrige, T. E. & Louis, A. A. Coarse-grained simulations of DNA overstretching. *J. Chem. Phys.* **138**, 085101 (2013).
7. Hatch, K., Danilowicz, C., Coljee, V. & Prentiss, M. Demonstration that the shear force required to separate short double-stranded DNA does not increase significantly with sequence length for sequences longer than 25 base pairs. *Phys. Rev. E* **78**, 011920 (2008).
8. Zou, Y. *et al.* A DNA Aptamer Targeting Cellular Fibronectin Rather Than Plasma Fibronectin for Bioimaging and Targeted Chemotherapy of Tumors. *Adv. Funct. Mater.* **32**, 2205002 (2022).

REVIEWER COMMENTS

Reviewer #3 (Remarks to the Author):

The authors properly addressed all my comments.

I recommend publication.

Reviewer #4 (Remarks to the Author):

Regarding comment 1: The force-induced drug uptake (DMC39pN) to target cells is evidenced (Fig. 3j). The force-induced drug uptake resulted in higher drug uptake into target cells than that into non-target cells (Fig. 3k and 3l). However, the non-specific cargo uptake (DMCrigid), not based on the force-induced mechanism, may also have relatively higher uptake into target cells than that into non-target cells although the absolute uptake level by the non-specific uptake is lower than that by the force-induced drug uptake mechanism.

Since both DMC39pN and DMCrigid have the RGD ligand, DMCrigid can interact with target cells as well as DMC39pN. If the authors want to show the force-induced drug uptake have higher specificity for drug delivery than non-specific cargo uptake, they need to examine whether the selectivity for GFP-Vin over Vin-KO (ca. 1.7- – 2-fold) of DMC39pN[Dex647N] is superior to that of DMCrigid[Dex647N]. I do not think that the authors have addressed this point in revised manuscripts.

Reviewer #3 (Remarks to the Author):

The authors properly addressed all my comments.

I recommend publication.

Reviewer #4 (Remarks to the Author):

Regarding comment 1: The force-induced drug uptake (DMC39pN) to target cells is evidenced (Fig. 3j). The force-induced drug uptake resulted in higher drug uptake into target cells than that into non-target cells (Fig. 3k and 3l). However, the non-specific cargo uptake (DMCrigid), not based on the force-induced mechanism, may also have relatively higher uptake into target cells than that into non-target cells although the absolute uptake level by the non-specific uptake is lower than that by the force-induced drug uptake mechanism. Since both DMC39pN and DMCrigid have the RGD ligand, DMCrigid can interact with target cells as well as DMC39pN. If the authors want to show the force-induced drug uptake have higher specificity for drug delivery than non-specific cargo uptake, they need to examine whether the selectivity for GFP-Vin over Vin-KO (ca. 1.7- – 2-fold) of DMC39pN[Dex647N] is superior to that of DMCrigid[Dex647N]. I do not think that the authors have addressed this point in revised manuscripts.

Response: We highly appreciate the careful critique of our manuscript as it has allowed us to more closely examine our work and strengthen it. In regard to this final comment, we very respectfully and kindly disagree with the reviewer's suggestion. We have already shown that the release of cargo is force-dependent rather than non-specific uptake with the data in **Fig. 3i, j**. We further demonstrated in **Fig. 3k, l** that vinculin knockdown results in reduced cargo uptake. This experiment was done with a mixed population (co-culture) of Vin(-) and Vin(+) cells and highlights force-specific uptake specifically into Vin(+) cells. We kindly disagree with the reviewer that DMC_{rigid} may have relatively higher uptake into target cells. Our data in fact shows that DMC_{rigid} has low uptake, even lower than DMC_{inert} (**Fig. 3i, j**). Furthermore, we had added in our last manuscript (pg. 9) revision that past literature indicates that different force-thresholds can alter the biology of cells. Specifically, cell migration or surface traversal is significantly reduced in high force-threshold surfaces such as DMC_{rigid} resulting in fewer probe interactions even within our experimental timeframes.¹

Force-mediated release is further strengthened in the manuscript with the *DMC_{39pN}[(HIF1 α)₂] and DMC_{39pN}[HIF1 α] data* as well with the *force-mediated uptake data in an in vivo mimicking environment* supplied in previous response to the reviewers. The exclusion of DMC_{rigid}[Dex647N] in Vin(-) and Vin(+) co-culture experiments is due to 1) DMC_{rigid} has been demonstrated to have lower release *already*, 2) DMC_{rigid} alters the biological behavior of cells and could potentially have confounding effects when used with a vinculin knockout cell line. Hence, we believe the manuscript has ample evidence to claim force-mediated uptake using the DMC system.

References

1. Sarkar, A., LeVine, D. N., Kuzmina, N., Zhao, Y. & Wang, X. Cell Migration Driven by Self-Generated Integrin Ligand Gradient on Ligand-Labile Surfaces. *Current Biology* **30**, 4022-4032.e5 (2020).

REVIEWER COMMENTS

Reviewer #4 (Remarks to the Author):

My question was how the selectivity for the target cell can be improved by the force-mediated uptake. But the authors responded with improved uptake efficiency, which is not the point.

I understand that the main focus of the manuscript is on the possibility of the force-mediated uptake of the nanostructure. However, since the authors mentioned its application to drug delivery, I wondered whether the force-mediated uptake of the structure is indeed beneficial for drug delivery, showing enhanced target cell selectivity compared to non-force-mediated uptake. Although the uptake efficiency in target cells is higher than that in non-target cells, enhanced target cell uptake may come with relatively increased uptake also into non-target cells. Therefore, to insist the potential of the force-mediated uptake in drug delivery, whether the selectivity of the force-mediated uptake for the target cell is improved compared to non-force-mediated uptake should be demonstrated.

Although I think that the authors have not addressed the critical comment explained above and thus that the manuscript is not ready for accept, I leave the decision with the editor and other reviewers.

Reviewer #4 (Remarks to the Author):

My question was how the selectivity for the target cell can be improved by the force-mediated uptake. But the authors responded with improved uptake efficiency, which is not the point.

I understand that the main focus of the manuscript is on the possibility of the force-mediated uptake of the nanostructure. However, since the authors mentioned its application to drug delivery, I wondered whether the force-mediated uptake of the structure is indeed beneficial for drug delivery, showing enhanced target cell selectivity compared to non-force-mediated uptake. Although the uptake efficiency in target cells is higher than that in non-target cells, enhanced target cell uptake may come with relatively increased uptake also into non-target cells. Therefore, to insist the potential of the force-mediated uptake in drug delivery, whether the selectivity of the force-mediated uptake for the target cell is improved compared to non-force-mediated uptake should be demonstrated. Although I think that the authors have not addressed the critical comment explained above and thus that the manuscript is not ready for accept, I leave the decision with the editor and other reviewers

Response: We understand the reviewers' concern about the target cell selectivity compared to non-force-mediated uptake. To clarify our perspective on this issue, we have summarized the experimental data below

1. DMCs can release dextran based on pN forces: To prove that DMCs release cargo specifically under force, we used DMC_{39pN} – which is a force-responsive mechanocapsule. As a control for the DMC_{39pN} , we created nearly identical DMCs that are non-force responsive (both inert and rigid). These were formulated as binary mixtures to equalize cRGD, DMC and Dextran densities on the surface. This experimental design produces controls with exact chemical composition, differing only in the force-thresholds of a fraction of the DMCs. Our data indeed shows that DMC_{39pN} has 2-fold higher uptake than DMC_{inert} and 3-fold higher uptake than DMC_{rigid} , indicating that dextrans inside force-sensitive DMCs are uptaken. *We attribute the lower uptake of DMC_{rigid} to the enhancement in cell migration on low force-threshold surfaces, which would lead to greater DMC encounter by cells and leading to increased uptake as reported in the past literature. This observation led us to drop DMC_{rigid} from further experiments as it alters the biology and the current data shows lower uptake even against DMC_{inert} .* Furthermore, comparing the uptake from the binary mixture of DMC_{39pN} [Dex_{647N}] + DMC_{inert} to the mixture DMC_{39pN} + DMC_{inert} [Dex_{647N}] mixture would be the most appropriate control.

2. DMCs can release cargo on fibronectin surfaces: To prove that DMCs can function outside glass substrates and in an *in vivo* setting we added aptamers to our DMCs for anchoring to fibronectin surfaces. This experiment is simpler without involving binary mixtures or additional DMC_{39pN} and allowing us to directly observe the fate of encapsulated dextrans in DMCs. Our data shows a 2-fold enhanced uptake from DMC_{39pN} over DMC_{inert} further demonstrating that DNA mechanocapsules can force-specifically release cargo. Using DMC_{rigid} as a control here would lead to the dissociation of the aptamers from fibronectin and releasing the whole DMC structure rendering it as an unsuitable control.

3. DMCs are selectively delivered to cells that generate greater traction forces in a mixed population: DMC_{39pN} along with DMC_{inert} were formulated as binary mixtures and two cell types were tested for uptake. We observe that a co-culture of Vinculin KO and Vin-GFP MEF cells on the DMC_{39pN} [Dex_{647N}] surface produces 2-fold differential uptake (Fig 3I) almost similar to the selectivity observed when they were cultured in isolation on 2 separate DMC_{39pN} [Dex_{647N}] surfaces (SI Fig 9E).

4. DMCs can deliver drugs based on forces over extended periods: To prove that DMCs can greatly enhance the selectivity of drug uptake and activity we formulated DMCs with established therapeutics. After 24 hours, we observed that the cells cultured on substrate with force-induced ASO delivery have a 20% downregulation of the target mRNA while the non-specific uptake remained not significant. The downregulation was scalable in that increasing the stoichiometry of the ASO to 2x inside the DMC doubled mRNA downregulation (Fig 4E). We would like to point out that conventional ASO treatment would conservatively require a 1000-fold greater amount of the drug than with our DMCs (see SI Fig 10).

We hope that the reviewer takes in to account the multiple lines of evidence that we have summarized here and understands our perspective that the data demonstrates not only relative uptake efficiency compared to DMC controls but also that the DMCs offer enhanced selectivity over conventional delivery.